# Laminin signals initiate the reciprocal loop that informs breast-specific gene expression and homeostasis by activating NO, p53 and microRNAs

Saori Furuta[1,2†*], Gang Ren[2], Jian-Hua Mao[1], Mina J Bissell[1*]

[1]Division of Biological Systems and Engineering, Lawrence Berkeley National Laboratory, Berkeley, United States; [2]Department of Cancer Biology, College of Medicine & Life Sciences, University of Toledo Health Science Campus, Toledo, United States

*For correspondence:
saori.furuta@utoledo.edu (SF);
MJBissell@lbl.gov (MJB)

Present address: [†]Department of Cancer Biology, College of Medicine & Life Sciences, University of Toledo Health Science Campus, Toledo, OH, United States

Competing interests: The authors declare that no competing interests exist.

**Abstract** How mammalian tissues maintain their architecture and tissue-specificity is poorly understood. Previously, we documented both the indispensable role of the extracellular matrix (ECM) protein, laminin-111 (LN1), in the formation of normal breast acini, and the phenotypic reversion of cancer cells to acini-like structures in 3-dimensional (3D) gels with inhibitors of oncogenic pathways. Here, we asked how laminin (LN) proteins integrate the signaling pathways necessary for morphogenesis. We report a surprising reciprocal circuitry comprising positive players: laminin-5 (LN5), nitric oxide (NO), p53, HOXD10 and three microRNAs (miRNAs) — that are involved in the formation of mammary acini in 3D. Significantly, cancer cells on either 2-dimensional (2D) or 3D and non-malignant cells on 2D plastic do not produce NO and upregulate negative players: NFκB, EIF5A2, SCA1 and MMP-9 — that disrupt the network. Introducing exogenous NO, LN5 or individual miRNAs to cancer cells reintegrates these pathways and induces phenotypic reversion in 3D. These findings uncover the essential elements of breast epithelial architecture, where the balance between positive- and negative-players leads to homeostasis.
DOI: https://doi.org/10.7554/eLife.26148.001

## Introduction

p53 is an extensively characterized regulator of gene expression in the context of malignant transformation and is aberrant in almost all cancer types. Many p53 studies have been performed in cells cultured in 2D conditions. Despite the extensive literature on p53 and its myriad of functions, little is known about what regulates p53 activity in higher organisms *in vivo* or about how p53 might regulate physiological tissue functions in 3D cultures (*Barcellos-Hoff et al., 1989*; *Petersen et al., 1992*; *Bissell et al., 2005*; *Lee et al., 2007*). ECM proteins, in particular LNs (*Miner and Yurchenco, 2004*), compose another important class of regulators that play a role in glandular tissue morphogenesis. Whether or how these two crucial regulators of gene expression intersect in tissue morphogenesis and homeostasis has not been examined.

To explore the possibility of such an interaction as an element of tissue-specificity, we utilized the HMT3522 cancer progression series of human mammary epithelial cells (MECs) (*Briand et al., 1987*; *Briand et al., 1996*; *Rizki et al., 2008*). This unique series comprise both primary normal epithelial cells or non-malignant cells (S1) derived from reduction mammoplasty, and their malignant counterpart (T4-2), which were derived without external oncogenic agents after prolonged cultivation in defined medium that lacked epidermal growth factor (EGF), followed by xenografts in animals (*Briand et al., 1987*). Non-malignant and malignant MECs and organoids are readily distinguished

**eLife digest** Most animal cells can secrete molecules into their surroundings to form a supportive meshwork of large proteins, called the extracellular matrix. This matrix is connected to the cell membrane through receptors that can transmit signals to the cell nucleus to change the levels of small RNA molecules called microRNAs. These, in turn, can switch genes on and off in the nucleus.

In the laboratory, cells that build breast tissue and glands can be grown in gels containing extracellular matrix proteins called laminins. Under these conditions, 'normal' cells form organized clusters that resemble breast glands. However, if the communication between healthy cells and the extracellular matrix is interrupted, the cells can become disorganized and start to form clumps that resemble tumors, and if injected into mice, can form tumors. Conversely, if the interaction between the extracellular matrix and the cells is restored, each single cancer cell can – despite mutations – be turned into a healthy-looking cell. These cells form a normal-looking tissue through a process called reversion. Until now, it was not known which signals help normal breast tissue to form, and how cancerous cells revert into a 'normal' shape.

To investigate this, Furuta et al. used a unique series of breast cells from a woman who underwent breast reduction. The cells taken from the discarded tissue had been previously grown by a different group of researchers in a specific way to ensure that both normal and eventual cancer cells were from the same individual. Furuta et al. then put these cells in the type of laminin found in extracellular matrix. The other set of cells used consisted of the same cancerous cells that had been reverted to normal-looking cells.

Analysis of the three cell sets identified 60 genes that were turned down in reverted cancer cells to a level found in healthy cells, as well as 10 microRNAs that potentially target these 60 genes. A database search suggested that three of these microRNAs, which are absent in cancer cells, are necessary for healthy breast cells to form organized structures. Using this as a starting point, Furuta et al. discovered a signaling loop that was previously unknown and that organizes breast cells into healthy looking tissue.

This showed that laminins help to produce nitric oxide, an important signaling molecule that activates several specific proteins inside the breast cells and restores the levels of the three microRNAs. These, in turn, switch off two genes that are responsible for activating an enzyme that can chop the laminins. Since the two genes are deactivated in the reverted cancer cells, the laminins remain intact and the cells can form organized structures. These findings suggest that if any of the components of the loop were missing, the cells would start to form cancerous clumps again. Reverting the cancer cells in the presence of laminins, however, could help cancer cells to form 'normal' structures again.

These findings shed new light on how the extracellular matrix communicates with proteins in the nucleus to influence how single cells form breast tissues. It also shows that laminins are crucial for generating signals that regulate both form and function of specific tissues. A better understanding of how healthy and cancerous tissues form and re-form may in the future help to develop new cancer treatments.

DOI: https://doi.org/10.7554/eLife.26148.002

by their colony structures in 3D LN1-rich ECM gels (lrECM) (*Petersen et al., 1992*). Non-malignant mammary cells form polarized colonies resembling normal acini of the breast (*Barcellos-Hoff et al., 1989*), whereas malignant cells form disorganized, tumor-like structures (*Petersen et al., 1992*; *Lee et al., 2007*; *Rizki et al., 2008*). However, if the architecture of colonies is restored in LN1 gels by downmodulating receptors such as integrins and EGFR, or other involved oncogenic pathways to a level found in normal cells, every single malignant cell would form polarized growth-arrested colonies – by a process we call phenotypic reversion – through a novel movement we have termed 'coherent angular motion' (CAMo) (*Tanner et al., 2012*).

Here, we aimed to delineate *core* regulators of proper ECM-chromatin communications that establish normal breast acinar architecture, a feature that is aberrant in cancer cells in 3D. Using S1 cells, T4-2 cells and T4-2 cells reverted to 'normal' phenotype (T4-2 Rev) by five different

signaling inhibitors, we identified a subset of 60 genes that had similar expression patterns in S1 and in all of the T4-2 Rev cells (*Bissell et al., 2005*; *Becker-Weimann et al., 2013*), as well as 10 miRNAs that could potentially target these 60 genes. Among the 10 miRNAs, we specifically focused on miR-34c-5p, −30e, and −144, which are dramatically downmodulated in many breast tumors (*Lu et al., 2005*).

Restoration of the miRNA caused phenotypic reversion of T4-2 cells in lrECM. While studying the signaling cascades that involve these three miRNAs, we identified a reciprocal regulatory network – comprising LN1 and LN5, NO, p53, HOXD10, NFκB, the three miRNAs, EIF5A2, SCA1, and MMP-9 – which connects the ECM-laminins and the nuclear transcription factors (TFs), most possibly via a newly discovered nuclear tunnel (*Jorgens et al., 2017*), to execute breast morphogenetic programs. Our results shed light on a completely novel and intricate reciprocal loop for breast acinar morphogenesis through a reiterative activation and suppression of regulatory molecules necessary to maintain the differentiated state in 3D and to prevent malignant conversion.

## Results

### Identification of miRNAs involved in the formation of mammary acini

Non-malignant S1 cells form apico-basally polarized acini in lrECM while conversely, malignant T4-2 cells form disorganized colonies (*Petersen et al., 1992*). We showed initially that inhibitory antibodies to beta-1 integrin reverted the malignant cells to 'normal' phenotype (*Figure 1a*) (*Weaver et al., 1997*). Inhibiting any of a dozen different oncogenic pathway components, including EGFR, PI3K and MMP-9, could revert breast cancer cells (*Figure 1a–1c*) (*Bissell et al., 2005*; *Beliveau et al., 2010*; *Becker-Weimann et al., 2013*). Such cross-modulation suggested the existence of central common integrators. Array analyses of the five most prominent reverting pathways identified 60 genes that were low in S1, and co-downregulated in T4-2 Rev cells (*Figure 1d*, *Table 1*) (*Bissell et al., 2005*), leading us to suspect that the common regulators would be miRNAs.

miRNA expression profiling of the S1, T4-2, and T4-2 Rev cells in lrECM identified a list of 30 miRNAs, the expression of which was anti-correlated with that of the 60 genes (*Figure 1d*, *Table 2*). Using a miRNA target database (microRNA.org), we predicted miRNAs that could potentially target the 60 genes. By combining these two lists, we chose 10 validated miRNAs (*Figure 2a*) each of which could potentially target at least 10 out of the 60 genes (*Table 3*).

Using published patient sample analyses, we selected three miRNAs: miR-34c-5p, −30e and −144, that were found to be downmodulated significantly in breast tumors and tumor cell lines (*Figure 1—figure supplement 1*)(GSE25464) (*Lu et al., 2005*). By *in situ* hybridization of tissue arrays containing 40 breast tumors vs. normal tissues, we confirmed a significant reduction of the three miRNAs in tumors (*Figure 2b, c*). Re-expression of each of the three miRNAs in T4-2 cells led to dramatic growth inhibition in soft agar (*Figure 2d*, *Figure 2—figure supplement 1b*) and caused phenotypic reversion in lrECM (*Figure 2e*, *Figure 2—figure supplement 1c*). Introduction of each of the three miRNAs into metastatic MDA-MB-231 breast cancer cells also led to severe growth impairment in lrECM (*Figure 2f*, *Figure 2—figure supplement 1d*). These results suggest that the three miRNAs are involved in inhibiting tumor cell growth and, by implication, in the maintenance of non-malignant cell behavior.

### EIF5A2 and SCA1 are the targets of the three identified miRNAs

A search of the miRNA target database (microRNA.org) identified EIF5A2 and SCA1 as the only common target genes of the three miRNAs among the 60 genes that were modulated by each of five reverting agents (*Table 4*, *Figure 1*). To validate this, we performed RT-PCR for EIF5A2 and SCA1 in T4-2 cells before and after miRNA expression. Endogenous levels of the two proteins were high in T4-2 cells compared to those in S1 cells, but as expected, were downmodulated in T4-2 Rev cells that were reverted either with a reverting agent or upon restoration of any of the three miRNAs (*Figure 3a*). Thus, each miRNA acted like a reverting agent, similar to the five other reverting agents we have reported on previously (*Figure 3a*; *Figure 1a–c*) (*Bissell et al., 2005*; *Beliveau et al., 2010*; *Becker-Weimann et al., 2013*). Importantly, depletion of either EIF5A2 or SCA1 in T4-2 cells with shRNA (*Figure 3—figure supplement 1a*) also caused phenotypic reversion (*Figure 3b*, *Figure 3—figure supplement 1b*). To ensure that this is not an off-target effect, we

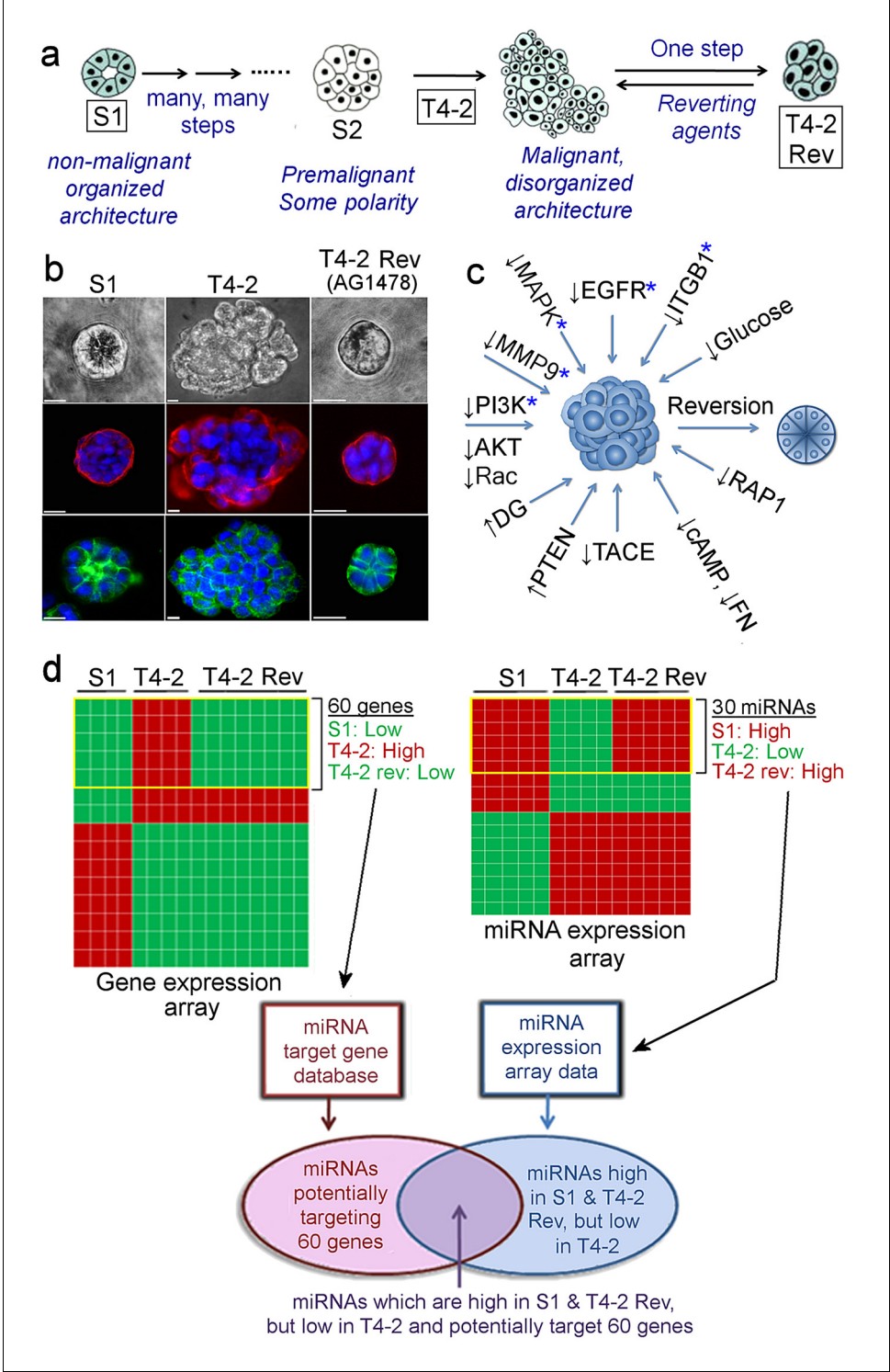

**Figure 1.** Identification of miRNAs linked to phenotypic reversion of human breast-cancer cells. (**a**) Scheme of progression of non-malignant HMT3522-S1 cells to malignant T4-2 cells and of reversion of T4-2 cells to an S1-like phenotype in the presence of a reverting agent. (**b**) S1, T4-2 and T4-2 Rev cells with AG1478 in lrECM. Cells are stained for integrin α6 (red), β-catenin (green) and nuclei (blue). Scale bars: 20 μm. Replicate experiments (n = 3) were performed, and representative data are shown. (**c**) A scheme of modulation of a single oncogenic pathway for phenotypic reversion of tumor cells. Five pathways chosen for gene and miRNA arrays are indicated with blue asterisks (*). (**d**) Screening miRNAs linked to phenotypic reversion. (Top left) Gene arrays (n = 5, GSE50444 [*Becker-Weimann et al., 2013*]) clustered 60 genes that are downmodulated in S1 and T4-2 Rev

*Figure 1 continued on next page*

*Figure 1 continued*

cells compared to T4-2 cells. (Top right) miRNA arrays (n = 4) clustered 30 miRNAs the expression of which was anti-correlated to that of these 60 genes. (Bottom left) A miRNA target database (microRNA.org) predicted miRNAs that could target the 60 genes. Combination of the two lists identified miRNAs that are linked to phenotypic reversion.

DOI: https://doi.org/10.7554/eLife.26148.003

The following figure supplement is available for figure 1:

**Figure supplement 1.** miR-34c, miR-30e and miR-144 are downregulated in breast cancer.

DOI: https://doi.org/10.7554/eLife.26148.004

---

restored EIF5A2 and SCA1 in T4-2 cells that were overexpressing the miRNAs. In these T4-2 cells we overexpressed cDNAs of EIF5A2 or SCA1 that lacked miRNA binding sites because the three miR-NAs bind only to the 3'UTR of the two target genes (*Table 5*). Overexpression was confirmed by western analysis (*Figure 3—figure supplement 1c*). Restoration of EIF5A2 or SCA1 severely impaired tumor-cell reversion, validating the importance of the inactivation of these two target genes for normal functional differentiation of breast acini (*Figure 3—figure supplement 1d and e*). These results demonstrate that the miRNA database correctly predicted EIF5A2 or SCA1 as the target genes of the three miRNAs.

## Reversion of breast tumor cells to normal phenotype requires upregulation of HOXD10 and downregulation of NFκB

To determine the regulators of the three miRNAs, we generated reporter constructs in which the luciferase gene was fused to the miRNA gene promoters, containing 3–0, 2–0 and 1–0 kb regions from the transcription start site (*Figure 3—figure supplement 2a*). The activity of the 1–0 kb region for miR-34c and the 3–0 kb region for both miR-30e and −144 was high in S1 and T4-2 Rev cells, but not in T4-2 cells (*Figure 3c*). In addition, we generated reporter constructs containing non-overlapping 3–2, 2–1 and 1–0 kb fragments of the miRNA promoters from the transcription start site (*Figure 3—figure supplement 2b*). The activity of the 1–0 kb region for miR-34c and the 3–2 kb region for both miR-30e and −144 was high in S1 and T4-2 Rev cells (*Figure 3d*).

To determine which TFs bound to these critical regions, we analyzed the PROMO database (*Farré et al., 2003*) and identified multiple high-confidence binding sites for HOXD10 and NFκB (% dissimilarity <15%; genomic frequency <$1 \times 10^{-4}$) (*Figure 4—figure supplement 1a*, *Table 6*) (*Farré et al., 2003*). We had shown previously that overexpression of HOXD10 or downmodulation of NFκB phenotypically reverts T4-2 cells (*Becker-Weimann et al., 2013*; *Chen et al., 2009*). As predicted, HOXD10 was high in S1 and T4-2 Rev cells compared to T4-2 cells (*Figure 4a*). By contrast, activation of NFκB, as measured by Ser536 phosphorylation of the p65 subunit that causes its nuclear translocation (*Sasaki et al., 2005*), was elevated in T4-2 cells and downmodulated in S1 and T4-2 Rev cells (*Figure 4a*). To show that these two TFs regulate the miRNAs in opposite directions, we generated T4-2 cells that were depleted of either p65 or p50, and its unprocessed precursor, p100, a subunit of NFκB. We also overexpressed HOXD10 in T4-2 cells (*Figure 4—figure supplement 1b*). In all these conditions, the activity of the miRNA promoters was elevated in the same regions as those described above (*Figure 4b*, *Figure 3—figure supplement 1d*). Northern analysis confirmed the increase of miRNA expression, allowing the formation of basally polarized colonies in lrECM (*Figure 4c and d*, *Figure 4—figure supplement 1c*), which were analogous to colonies of miRNAs-expressing T4-2 cells (*Figure 2c*). These results highlight the importance of the ratios and balance of different regulatory genes in maintaining normal architecture.

To prove that HOXD10 and NFκB do indeed bind the promoters of the three miRNAs, we performed chromatin immunoprecipitation (ChIP) analyses. We found that HOXD10 bound the promoters of the three miRNAs in S1 and T4-2 Rev cells, but not in T4-2 cells, whereas the NFκB p65 subunit bound the same regions in T4-2 cells, but not in S1 and T4-2 Rev cells (*Figure 4e*, *Figure 4—figure supplement 1d*).

To ascertain the functional consequence of the above experiment, we used the decoy technology described by Osako et al. (*Osako et al., 2012*). These decoys were derived from their respective binding sequences in each miRNA promoter (*Table 7*). For T4-2 cells, which have a high level of endogenous NFκB (*Figure 4a*), we expressed NFκB decoys; for T4-2 cells that we overexpressed

**Table 1.** List of 60 genes downregulated in T4-2 revertants to the level found in S1 but unmodulated in T4-2 cells in lrECM (p-value<0.05 was considered significant) (*Rizki et al., 2008*).

| Ensembl gene ID | Ensembl transcript ID | Gene name |
| --- | --- | --- |
| ENSG00000107796 | ENST00000224784 | ACTA2 |
| ENSG00000109321 | ENST00000264487 | AREG |
| ENSG00000102606 | ENST00000317133 | ARHGEF7 |
| ENSG00000134107 | ENST00000256495 | BHLHB2 |
| ENSG00000101189 | ENST00000217161 | C20ORF20 |
| ENSG00000115009 | ENST00000358813 | CCL20 |
| ENSG00000161570 | ENST00000293272 | CCL5 |
| ENSG00000169583 | ENST00000224152 | CLIC3 |
| ENSG00000165959 | ENST00000298912 | CLMN |
| ENSG00000176390 | ENST00000324238 | CRLF3 |
| ENSG00000105246 | ENST00000221847 | EBI3 |
| ENSG00000163577 | ENST00000295822 | EIF5A2 |
| ENSG00000187266 | ENST00000222139 | EPOR |
| ENSG00000085832 | ENST00000262674 | EPS15 |
| ENSG00000124882 | ENST00000244869 | EREG |
| ENSG00000197930 | ENST00000359133 | ERO1L |
| ENSG00000149573 | ENST00000278937 | EVA1 |
| ENSG00000141524 | ENST00000322933 | EVER1 |
| ENSG00000185862 | ENST00000330927 | EVI2B |
| ENSG00000180263 | ENST00000343958 | FGD6 |
| ENSG00000088726 | ENST00000314124 | FLJ11036 |
| ENSG00000137312 | ENST00000259846 | FLOT1 |
| ENSG00000100031 | ENST00000248923 | GGT1 |
| ENSG00000149435 | ENST00000286890 | GGTLA4 |
| ENSG00000051620 | ENST00000058691 | HEBP2 |
| ENSG00000178922 | ENST00000326220 | HT036 |
| ENSG00000172183 | ENST00000306072 | ISG20 |
| ENSG00000105655 | ENST00000357050 | ISYNA1 |
| ENSG00000119698 | ENST00000304338 | KIAA1622 |
| ENSG00000134121 | ENST00000256509 | L1CAM |
| ENSG00000110492 | ENST00000359803 | MDK |
| ENSG00000146232 | ENST00000275015 | NFKBIE |
| ENSG00000008517 | ENST00000008180 | NK4 |
| ENSG00000157045 | ENST00000287706 | NTAN1 |
| ENSG00000135124 | ENST00000356268 | P2R $\times$ 4 |
| ENSG00000110218 | ENST00000227638 | PANX1 |
| ENSG00000145431 | ENST00000274071 | PDGFC |
| ENSG00000166289 | ENST00000299373 | PLEKHF1 |
| ENSG00000083444 | ENST00000196061 | PLOD |
| ENSG00000107758 | ENST00000342558 | PPP3CB |
| ENSG00000011304 | ENST00000350092 | PTBP1 |
| ENSG00000073756 | ENST00000186982 | PTGS2 |
| ENSG00000118508 | ENST00000237295 | RAB32 |
| ENSG00000013588 | ENST00000014914 | RAI3 |

*Table 1 continued on next page*

*Table 1 continued*

| Ensembl gene ID | Ensembl transcript ID | Gene name |
|---|---|---|
| ENSG00000168501 | ENST00000307470 | RDBP |
| ENSG00000136643 | ENST00000259161 | RPS6KC1 |
| ENSG00000124788 | ENST00000244769 | SCA1 |
| ENSG00000181788 | ENST00000312960 | SIAH2 |
| ENSG00000136603 | ENST00000259119 | SKIL |
| ENSG00000173262 | ENST00000340749 | SLC2A14 |
| ENSG00000059804 | ENST00000075120 | SLC2A3 |
| ENSG00000160326 | ENST00000291725 | SLC2A6 |
| ENSG00000086300 | ENST00000338523 | SNX10 |
| ENSG00000061656 | ENST00000080856 | SPAG4 |
| ENSG00000141380 | ENST00000269137 | SS18 |
| ENSG00000198203 | ENST00000251481 | SULT1C1 |
| ENSG00000152284 | ENST00000282111 | TCF7L1 |
| ENSG00000035862 | ENST00000262768 | TIMP2 |
| ENSG00000125657 | ENST00000245817 | TNFSF9 |
| ENSG00000115652 | ENST00000283148 | UXS1 |

DOI: https://doi.org/10.7554/eLife.26148.007

HOXD10 (*Figure 4—figure supplement 1b*), we employed HOXD10 decoys. Any alteration in the promoter activity after expressing a particular decoy would indicate that the TF was bound and sequestered by the decoy. To test for sequence-specific binding of the TFs, we engineered decoys harboring point mutations in T4-2 cells. The expression of wild-type NFκB decoys, but not mutant decoys, derepressed the promoter activities, showing that the wild-type decoys bound and sequestered NFκB, whereas the mutant decoys did not. The procedure was repeated for HOXD10 with similar conclusions (*Figure 4—figure supplement 1e*). Collectively, these results demonstrate that HOXD10 and NFκB directly bind the specific sequences in miRNA promoters in a mutually exclusive manner to regulate miRNA expression for restoration of breast acinar architecture.

## p53 is another essential element in mammary acinar formation and tumor-cell reversion

p53 is a potent inhibitor of NFκB (*Webster and Perkins, 1999*; *Murphy et al., 2011*). Because p53 activity in tumors is extremely high, it is often assumed that little or no p53 is present in normal tissues. We found appreciable levels of wild-type p53 in the epithelial compartment of sections of normal breast tissues but not in the stroma (*Figure 9—figure supplement 1*). In 3D cultures of S1 and T4-2 Rev cells, we found appreciable levels of Ser20-phosphorylated p53 (pSer20-p53), which stabilizes (*Chehab et al., 1999*) and enhances the transactivation activity of p53 (*Jabbur et al., 2000*). This was also the case when either of the miRNAs were overexpressed in T4-2 cells or when their inhibitory target, EIF5A2 or SCA1, was depleted (*Figure 5a*). The expression of the p53-regulated genes, p21, GADD45 and DRAM, was elevated in S1 and all T4-2 Rev cells (*Figure 5a*).

Whether p53 is both the direct inhibitor of NFκB and an activator of HOXD10 was examined by overexpressing the dominant-negative p53 (DNp53) (*Harvey et al., 1995*) in S1 cells. This particular mutant of p53 was reported to effectively abolish tumor suppression and transcriptional activity of the endogenous wild-type p53, leading to enhanced tumor growth, even in heterozygous mice. In S1 cells that overexpressed DNp53, HOXD10 level plummeted as NFκB activity, measured by Ser536 phosphorylation of the p65 subunit, increased over the levels seen in control S1 cells or S1 cells overexpressing the wild-type p53 (*Figure 5b*).

As expected, expression of DNp53 prevented S1 cells from forming polarized quiescent acini in lrECM (*Figure 5c*, *Figure 5—figure supplement 1a*). Similarly, RNAi-mediated depletion of the wild-type p53 in S1 or MCF10A cells abrogated acinar formation (*Figure 5d and e*, *Figure 5—figure*

**Table 2.** List of 30 miRNAs that were upregulated in S1 and T4-2 revertants and downmodulated in T4-2 in lrECM (p-value<0.05 was considered significant).

| Mature ID | Fold regulation S1 vs T4-2 | Fold regulation T4 rev vs T4 | P value |
|---|---|---|---|
| miR-450b-5p | 30.3789 | 869.8262 | 0.049943 |
| miR-105 | 11.1967 | 783.9313 | 0.007486 |
| miR-383 | 52.5275 | 735.6709 | 0.042511 |
| miR-432 | 17.7736 | 541.6623 | 0.020574 |
| miR-495 | 455.6135 | 510.0813 | 0.004495 |
| miR-30e | 65.4581 | 228.7297 | 0.047957 |
| miR-190 | 48.2236 | 221.0022 | 0.044772 |
| miR-369–3 p | 14.1069 | 158.4536 | 0.041128 |
| miR-323–3 p | 13.8486 | 118.1588 | 0.015214 |
| miR-127–5 p | 10.5195 | 115.9948 | 0.005194 |
| miR-330–3 p | 39.2603 | 113.8705 | 0.044612 |
| miR-382 | 24.6754 | 82.0207 | 0.021385 |
| miR-337–3 p | 36.2104 | 35.2915 | 0.003663 |
| miR-423–3 p | 55.2984 | 32.9948 | 0.045694 |
| miR-125b | 48.925 | 29.9434 | 0.04939 |
| miR-376a | 212.9199 | 11.2032 | 0.049943 |
| miR-296–5 p | 42.5671 | 8.5618 | 0.045775 |
| miR-135a | 60.0253 | 7.5379 | 0.045617 |
| miR-144 | 1234.0342 | 7.3743 | 0.003973 |
| miR-301b | 32.1668 | 7.1892 | 0.043739 |
| miR-376c | 49.8377 | 6.2803 | 0.046834 |
| miR-487a | 68.5143 | 6.2243 | 0.035722 |
| miR-590–3 p | 14.4952 | 5.9622 | 0.035266 |
| miR-301a | 30.8564 | 5.4045 | 0.047854 |
| miR-98 | 33.6103 | 5.1012 | 0.041169 |
| miR-34c-5p | 31.215 | 4.8038 | 0.043702 |
| miR-496 | 42.8632 | 3.2108 | 0.044913 |
| miR-543 | 74.4569 | 2.897 | 0.01967 |
| miR-143 | 590.5164 | 2.2076 | 0.042274 |
| miR-374a | 11.1001 | 1.1926 | 0.013047 |

DOI: https://doi.org/10.7554/eLife.26148.008

*supplement 1b and c*). Furthermore, inhibition of p53 activity with a specific inhibitor, α-pifithrin (*Komarov et al., 1999*), rendered T4-2 cells resistant to phenotypic reversion by any of the reverting agents tested (*Bissell et al., 2005*; *Lee et al., 2007, 2012*) or by re-expression of any of the three miRNAs (*Figure 5f and g*, *Figure 5—figure supplement 1d and e*). Likewise, MCF10A cells that overexpressed DNp53, were resistant to reverting agents (*Figure 5h*, *Figure 5—figure supplement 1f*).

## *De novo* synthesized LN5 is required for acinar morphogenesis

It is known that the basement membrane (BM) of the mammary gland includes not only LN1 but also LN5. To maintain tissue architecture, signaling pathways need to regulate each other directly or indirectly (*Bissell et al., 1982*, *2005*). We had shown previously that even after placing cells in lrECM, formation of acini and production of milk proteins still required an endogenously formed BM (*Streuli and Bissell, 1990*). Accordingly, we measured the levels of human LNs in the conditioned media (CM) and in cell lysates of S1 and T4-2 cells grown in lrECM. Using a human-specific pan-LN

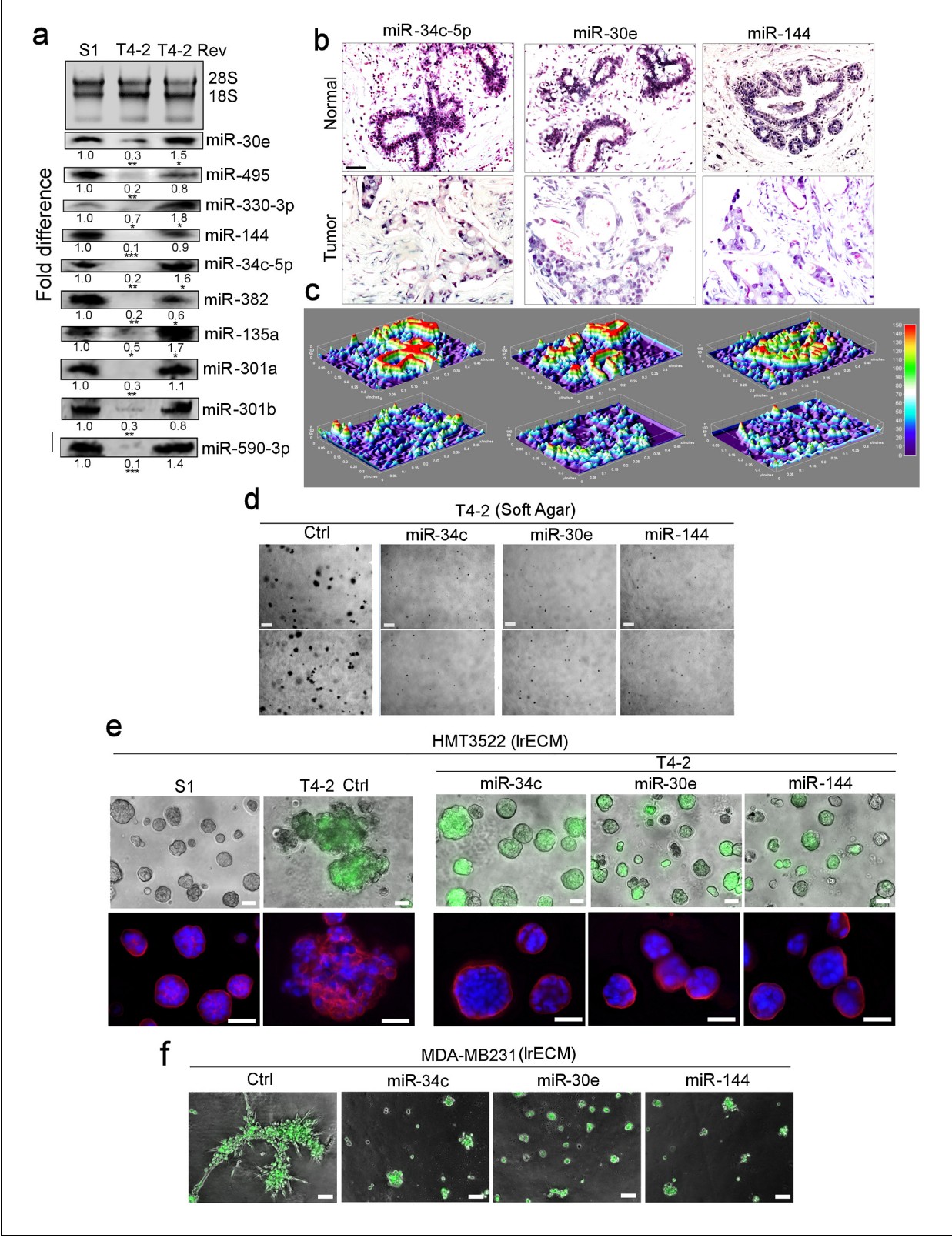

**Figure 2.** Restoring the expression of miR-34c-5p, −30e or −144 in breast cancer cells induces phenotypic reversion. (a) Actual expression pattern of the ten identified miRNAs (*Figure 1*): the levels are at least two-fold higher in S1 and T4-2 Rev cells than in T4-2 cells, as measured by northern analysis. 28S and 18S RNAs were used as internal controls. Fold difference was determined with respect to S1 cells. *p<0.05; **p<0.01; and ***p<0.001. (b) *In situ* hybridization of primary human breast tissues showed the abundance of miR-34c-5p, −30e and −144 in normal (top row) compared to tumor

*Figure 2 continued on next page*

*Figure 2 continued*

tissues (second row) (n = 3). Nuclei were counterstained with nuclear fast red. (**c**) Heat maps of (**b**) generated by ImageJ. (**d**) T4-2 cells expressing the 3 miRNAs grown in soft agar. See quantification in *Figure 2—figure supplement 1b*. Two representative images are shown out of 9 samples (**e**) T4-2 cells expressing thethree miRNAs grown in lrECM. (Top) Phase images overlaid with FITI to indicate transduced cells. (Bottom) Red: α6 integrin; blue: DAPI. See quantification in *Figure 2—figure supplement 1c*. (**f**) MDA-MB231 cells expressing the three miRNAs grown in lrECM. Phase images overlaid with FITI to indicate transduced cells. Scale bars: 20 μm. See quantification of colony sizes in *Figure 2—figure supplement 1d*. For each analysis, replicate experiments (n = 3) were performed, and representative data are shown.

DOI: https://doi.org/10.7554/eLife.26148.005

The following figure supplement is available for figure 2:

**Figure supplement 1.** miR-34c, miR-30e and miR-144 are critically involved in tumor-cell reversion.

DOI: https://doi.org/10.7554/eLife.26148.006

antibody, we observed a significant increase in human LNs in both CM and lysates of S1 and T4-2 Rev cells reverted by expression of miRNAs or depletion of the two target genes (*Figure 6—figure supplement 1a*).

Functional LN proteins are heterotrimers of αβγ chains (*Miner and Yurchenco, 2004*). To determine which LN trimers were upregulated, we analyzed the CM of cells grown in lrECM cultures using antibody arrays against human ECM proteins. The α3, β3 and γ2 chains of LN5 were highly elevated in S1 and T4-2 Rev cells that expressed the miRNAs or that were depleted of their two targets. By contrast, parental T4-2 cells did not produce LN5, suggesting that LN5 is only expressed in MECs capable of forming acinar-like polarized structures (*Figure 6—figure supplement 1b–d*). To test the possibility, we depleted one of the LN5 subunits, LAMA3, with shRNA. Loss of LAMA3 abrogated reversion of T4-2 cells with any of the different reverting agents including any of the three miRNAs (*Figure 6a and b*, *Figure 6—figure supplement 2a, b*). Depletion of LAMA3 could be rescued by addition of ectopic LN5, confirming the specificity of the reaction (*Figure 6a and b*, *Figure 6—figure supplement 2a and b*).

To follow how LN5 was elevated in acinar formation and tumor cell reversion, we postulated that it could be due to LN5 protein stabilization due to suppression of MMP-9 transcription. We previously had shown that MMP-9, a metalloproteinase secreted to degrade LNs, is elevated in T4-2, but downmodulated in T4-2 Rev cells, leading to stabilization of secreted LNs (*Beliveau et al., 2010*).

**Table 3.** List of 12 miRNAs that were upregulated in S1 and T4-2 revertants and downmodulated in T4-2, and that could target more than 10 genes among the 60 genes that showed the opposite expression patterns.

| Mature ID | Fold regulation T4 vs Control | Fold regulation T4 Rev vs T4 | p-value | # Targets / 60 genes | Gene locus | Type | |
|---|---|---|---|---|---|---|---|
| miR-450b-5p | −30.3789 | 869.8262 | 0.000304 | 15 | Xq26.3 | Intergenic | |
| miR-495 | −455.6135 | 510.0813 | 0.028964 | 12 | 14q32.31 | Intergenic | |
| **miR-30e*** | **−65.4581** | **228.7297** | **0.008983** | **11** | **1p34.2** | **Intronic** | Down (p<0.05) |
| miR-330–3 p | −39.2603 | 113.8705 | 0.007481 | 22 | 19q13.32 | Intronic | |
| miR-382 | −24.6754 | 82.0207 | 0.001901 | 12 | 14q32.31 | Intergenic | |
| miR-423–3 p | −55.2984 | 32.9948 | 0.011304 | 14 | 17q11.2 | Intronic | |
| miR-135a | −60.0253 | 7.5379 | 0.039409 | 13 | 3p21.1 | Intergenic | |
| | | | | | 12q23.1 | Intergenic | |
| **miR-144*** | **−1234.0342** | **7.3743** | **0.010599** | **12** | **17q11.2** | **Intergenic** | Down (p<0.05) |
| miR-301b | −32.1668 | 7.1892 | 0.028553 | 16 | 22q11.21 | Intergenic | |
| miR-590–3 p | −14.4952 | 5.9622 | 0.043351 | 21 | 7q11.23 | Intronic | |
| miR-301a | −30.8564 | 5.4045 | 0.044539 | 13 | 17q22 | Intronic | |
| **miR-34c-5p*** | **−31.215** | **4.8038** | **0.01567** | **10** | **11q23.1** | **Intergenic** | Down (0 < 0.05) |

*The three miRNAs in **bold** that were the focus of this study.

[†]p-values were obtained from the array results [GSE2564] (*Petersen et al., 1998*).

DOI: https://doi.org/10.7554/eLife.26148.009

**Table 4.** List of genes targeted by miR-34c-5p, miR-30e and miR-144 among the cluster of 60 genes that were downmodulated in S1 and T4-2 revertants and upregulated in T4-2.

| miR-144 (12 targets) | miR-30e (11 targets) | mir-34c-5p (10 targets) |
|---|---|---|
| | | AREG |
| | | BHLHB2 |
| C20ORF20 | | |
| CCL20 | | |
| EIF5A2 | EIF5A2 | EIF5A2 |
| EREG | | |
| ERO1L | | ERO1L |
| | FGD6 | |
| | | ISG20 |
| | | NK4 |
| | KIAA1622 | |
| L1CAM | L1CAM | |
| PTBP1 | | |
| PTGS2 | PTGS2 | |
| | RDBP | RDBP |
| SCA1 (ATXN1) | SCA1 (ATXN1) | SCA1 (ATXN1) |
| SIAH2 | | SIAH2 |
| | SLC2A14 | |
| | SLC2A3 | |
| | SNX10 | SNX10 |
| SS18 | SS18 | |
| TIMP2 | | |

DOI: https://doi.org/10.7554/eLife.26148.013

We measured the level of secreted MMP-9 in lrECM cultures and showed that MMP-9 was significantly reduced in T4-2 cells that expressed any of the three miRNAs- or were depleted of the two target genes, EIF5A2 and SCA1 (*Figure 6c*). It had been shown previously that both EIF5A2 and SCA1 lie downstream of the PI3K/AKT pathway and are involved in positive regulation of MMP transcription (*Liu et al., 2000*; *Park et al., 2013*; *Khosravi et al., 2014*). We concluded that expression of the miRNAs inactivates both EIF5A2 and SCA1 and thus downmodulates MMP-9 leading to stabilization of LN5.

## p53 activation during acinar formation is triggered by LN-induced nitric oxide (NO) production

We searched for possible explanations of how LNs activate p53. In older literature, LN was reported to induce NO production in neuronal and endothelial cells as part of mechanotransduction pathways (*Gloe and Pohl, 2002*; *Rialas et al., 2000*). As NO is reported to be a potent activator of p53 (*Forrester et al., 1996*; *Wang et al., 2002*), we hypothesized that LNs might also be instrumental in inducing NO production in breast cells, which in turn would activate p53. We applied purified LN5 (*Figure 7a*) or lrECM (*Figure 7—figure supplement 1a*) to MCF10A cells and observed an increase in pSer20-p53 after 30 min, along with increases in Ser1981-phosphorylated ATM and total level of p14 ARF, the known p53 activators (*Canman et al., 1998*; *Zhang et al., 1998*). Under the same conditions, Ser1417 phosphorylation of nitric oxide synthase 1 (NOS-1) was also elevated (*Figure 7a*, *Figure 7—figure supplement 1a*), suggesting its role in NO production. In contrast, DNp53 overexpressed in MCF10A cells was not activated in response to LNs, whereas ATM, p14 ARF and NOS-1 were all activated (*Figure 7a*, *Figure 7—figure supplement 1a*). When MCF10A cells were treated

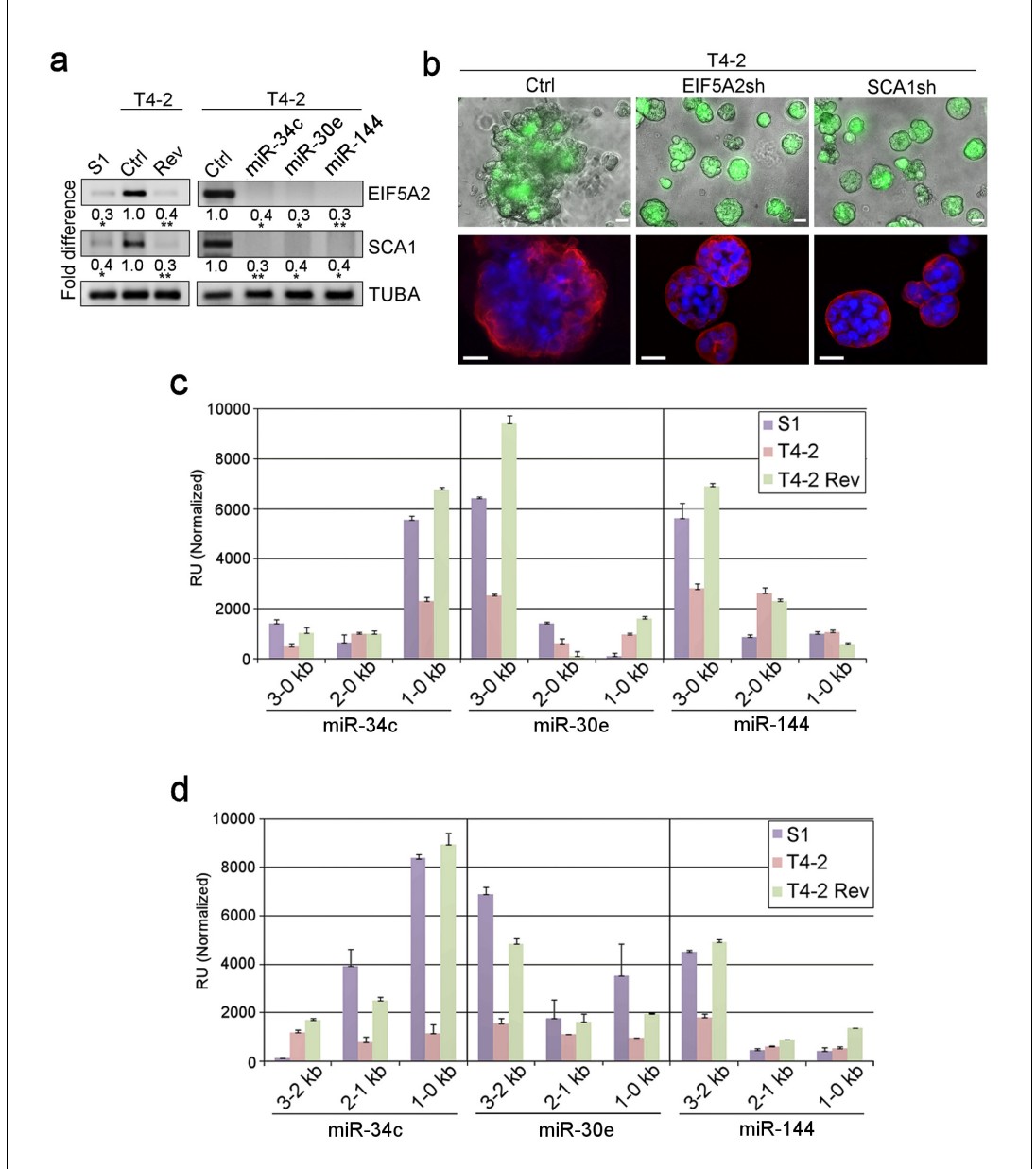

**Figure 3.** Dissection of miRNA target genes and promoter regulation. (a) Representative result of semi-quantitative RT-PCR (n = 3) to determine the levels of EIF5A2 and SCA1 in S1, T4-2 and T4-2 Rev cells (treated with AG1478). Fold difference was determined with respect to the Ctrl T4-2. *p<0.05 and **p<0.01 (b) T4-2 cells depleted of EIF5A2 or SCA1 grown in lrECM. (Top) Phase images overlaid with FITI to indicate transduced cells. (Bottom) Red — integrin α6 (red); blue — DAPI. Scale bars: 20 μm. See quantification in **Figure 3—figure supplement 1b**. (c) Activities of different miRNA promoters (n = 3) in S1, T4-2 and T4-2 Rev cells using the promoter constructs shown in **Figure 3—figure supplement 2a**. Note that a 1–0 kb fragment of miR-34c promoter, a 3–0 kb fragment of miR-30e promoter and a 3–0 kb fragment of miR-144 promoter were activated in the S1 and T4-2 Rev cells. (d) Activities of different miRNA promoters (n = 3) in S1, T4-2 and T4-2 Rev cells using the promoter constructs shown in **Figure 3—figure supplement 2b**. Note that a 1–0 kb fragment of miR-34c promoter,a 3–2 kb fragment of miR-30e promoter and a 3–2 kb fragment of miR-144 promoter were activated in S1 and T4-2 Rev cells. Data are represented as mean ± SEM. For each analysis, replicate experiments (n = 3) were performed, and representative data are shown.

DOI: https://doi.org/10.7554/eLife.26148.010

The following figure supplements are available for figure 3:

**Figure supplement 1.** Identification of the downstream targets of the three miRNAs and generation of the promoter constructs of the miRNA genes.
DOI: https://doi.org/10.7554/eLife.26148.011

**Figure supplement 2.** Scheme of reporter constructs derived from the promoter regions of the three miRNAs genes.

*Figure 3 continued on next page*

*Figure 3 continued*

DOI: https://doi.org/10.7554/eLife.26148.012

with a NOS inhibitor, L-NAME, that inhibits NO production, LN5-mediated activation of p53, as well as of ATM and p14 ARF, were severely impaired (*Figure 7b*).

We measured the level of NO in CM after addition of LNs using a fluorescence probe, DAN, against NO metabolites. S1 and MCF10A cells produced NO as a function of time in response to LN5 or lrECM (*Figure 7c and d*). By contrast, T4-2 cells failed to do so (*Figure 7c and d*). Addition of another ECM protein collagen-1 (COL1) did not induce NO production by S1 or MCF10A cells (*Figure 7e*), suggesting a unique role of LNs. We then monitored the intracellular NO level after addition of lrECM using a fluorescence probe DAF-FM DA. NO level peaked at around 1 hr after lrECM addition and declined thereafter in S1 and MCF10A cells, whereas it remained low in T4-2 cells (*Figure 7f*, *Figure 7—figure supplement 1b*).

To confirm the biological relevance of NO production by MECs, we stained 3D colonies for S-nitrosocysteine (SNOC), an indicator of NO production (*Gould et al., 2013*) and localization (*Iwakiri et al., 2006*). S1 acini showed strong basolateral SNOC staining, whereas T4-2 cells showed weak and dispersed staining. However, T4-2 Rev cells restored the strong basolateral SNOC staining analogous to S1, suggesting the recovery of NO production upon phenotypic reversion (*Figure 7g*, *Figure 7—figure supplement 1c*). We then stained normal (n = 8) vs. cancerous (n = 32) breast tissue sections for SNOC. Normal mammary epithelia were distinctively stained for SNOC, whereas the majority of tumor samples were only weakly and diffusely stained [positive staining (intensity >+1): 8/8 vs. 8/32, respectively] (*Figure 7h*). These results support the relevance of NO production to the biology of the normal breast.

## NO is critical for mammary acinar formation and gland morphogenesis

NO is known to play a role in the differentiation and morphogenesis of neurons, muscles and immune cells (*Rialas et al., 2000*; *Stamler and Meissner, 2001*; *Niedbala et al., 2002*). To test the involvement of NO in mammary morphogenesis, we inhibited NO production with L-NAME in two different non-malignant breast epithelial cells; this led to the formation of disorganized proliferative structures in lrECM (*Figure 8a and b*, *Figure 8—figure supplement 1a and b*). Alternatively, the induction of NO production in T4-2 cells with a NO donor, SNAP, induced phenotypic reversion (*Figure 8c*, *Figure 8—figure supplement 1c*). Also, application of L-NAME to T4-2 cells, even in the presence of a reverting agent (e.g., an inhibitor of EGFR or β1 integrin) (*Bissell et al., 2003*), abrogated phenotypic reversion in lrECM (data not shown). To determine whether the activity of NO is necessary for human mammary gland morphogenesis, we monitored the alveologenesis of breast organoids treated with L-NAME in *ex vivo* 3D cultures. L-NAME treatment dramatically reduced the percentage of colonies capable of alveologenesis (vehicle-treated: 28% vs. L-NAME-treated: 1.2%) (*Figure 8d*, *Videos 1* and *2*).

We tracked movement of L-NAME-treated S1 cells in lrECM for 48 hr by live cell imaging. We and others have shown previously that acinar forming non-malignant breast cells undergo CAMo in

**Table 5.** Predicted binding sites of three miRNAs at 3′UTR of SCA1 and EIF5A2.

| miRNA | Seed | miRNA binding site in 3′UTR of mRNA |
|---|---|---|
| **SCA1** | | |
| hsa-miR-34c | GGCAGUG | 691, 913, 6461 |
| hsa-miR-30e | GUAAACA | 3588, 4308, 4603, 4770, 5440, 6092, 6233 |
| hsa-miR-144 | GAUAUCA | 47, 987, 1113, 1160, 4557, 5426, 6267 |
| **EIF5A2** | | |
| hsa-miR-34c | GGCAGUG | 297, 802, 2656 |
| hsa-miR-30e | GUAAACA | 3264, 3312, 3525, 3652, 3643, 4113 |
| hsa-miR-144 | GAUAUCA | 2642, 2742, 2977, 4123, 4541, 4570 |

DOI: https://doi.org/10.7554/eLife.26148.014

**Table 6.** miRNA promoter regions harboring binding sites of TFs, NFκB (p65) and HOXD10.

| miRNA promoter | TXN bound | Start nt from TSS | End nt from TSS | String | Dissimilarity (%) | Frequency (random expectancy x $10^{-3}$) | |
|---|---|---|---|---|---|---|---|
| | | | | | | Equally | Query |
| miR-34c | RelA | −787 | −778 | AGGGAATCAA | 14 | $1 \times 10^{-5}$ | $1 \times 10^{-5}$ |
| | | −769 | −760 | TGGGAAGTTT | 11 | $3 \times 10^{-5}$ | $5 \times 10^{-5}$ |
| | | −427 | −418 | TGGGAACCTT | 11 | $4 \times 10^{-5}$ | $3 \times 10^{-5}$ |
| | | −64 | −55 | TGGGAAGCCG | 13 | $4 \times 10^{-5}$ | $4 \times 10^{-5}$ |
| | | −56 | −47 | CGCTTTCCCA | 12 | $5 \times 10^{-5}$ | $4 \times 10^{-5}$ |
| | | -3 | 6 | GGGGAATGAG | 13 | $3 \times 10^{-5}$ | $3 \times 10^{-5}$ |
| | HOXD10 | −864 | −855 | AGTTTGTATT | 10 | $1 \times 10^{-4}$ | $2 \times 10^{-4}$ |
| | | −385 | −376 | CCCTTCTATT | 12 | $3 \times 10^{-5}$ | $4 \times 10^{-5}$ |
| miR-30e | RelA | −2888 | −2879 | GATATTCCCA | 2 | $6 \times 10^{-6}$ | $5 \times 10^{-6}$ |
| | HOXD10 | −2553 | −2544 | TGGTTGTATT | 10 | $1 \times 10^{-4}$ | $2 \times 10^{-4}$ |
| | | −2230 | −2221 | GCGTGATATT | 11 | $1 \times 10^{-4}$ | $1 \times 10^{-4}$ |
| | | −2122 | −2113 | TTTTTTTATT | 4 | $1 \times 10^{-5}$ | $4 \times 10^{-5}$ |
| | | −1967 | −1967 | TACTCATATT | 9 | $1 \times 10^{-4}$ | $2 \times 10^{-4}$ |
| miR-144 | RelA | −3207 | −3198 | AGGGAATTTG | 10 | $5 \times 10^{-5}$ | $5 \times 10^{-5}$ |
| | HOXD10 | −2850 | −2841 | AATAGAATGA | 10 | $1 \times 10^{-4}$ | $2 \times 10^{-4}$ |
| | | −2715 | −2706 | AATACAAAAA | 10 | $1 \times 10^{-4}$ | $2 \times 10^{-4}$ |
| | | −2412 | −2403 | CCATATTATT | 11 | $1 \times 10^{-4}$ | $1 \times 10^{-4}$ |
| | | −2336 | −2327 | AATAAGAGTA | 7 | $5 \times 10^{-5}$ | $7 \times 10^{-5}$ |
| | | −2312 | −2303 | ATTTATTATT | 10 | $1 \times 10^{-4}$ | $2 \times 10^{-4}$ |

DOI: https://doi.org/10.7554/eLife.26148.017

lrECM, whereas cancer cells exhibit random amoeboid motion (*Tanner et al., 2012*; *Wang et al., 2013*). S1 cells treated with L-NAME are defective in CAMo and form disorganized masses (*Figure 8e*, *Videos 3* and *4*).

## LN5 activates p53 phosphorylation and p53 activates LN5 transcription

We showed above that NO production in response to lrECM is critical for p53 activation and the formation of mammary acini (*Figures 7a–g* and *8a–e*). This process involves *de novo* synthesized LN5 (*Figure 6a and b*). We also showed that p53 upregulates the expression of HOXD10 and down-regulates activation of NFκB. This dual action allows expression of the three miRNAs that inhibit TFs, SCAI and EIF5A2, to downmodulate MMP-9 expression. The result is inhibition of laminin protein degradation, leading to the closure of the morphogenetic loop (*Figure 9a*).

To demonstrate reciprocity in 3D, we selected the interaction between p53 and LN5, where a single manipulation at any part of the cycle allowed integration of all the pathways examined, PROMO analysis of the promoter of *LAMA3* chain of LN5 revealed over 20 high-confidence p53 binding sites within 1 kb length of the CpG island around the transcription start site (% dissimilarity <8%; genomic frequency <$1\times10^{-3}$) (*Figure 9b*, *Table 8*) (*Farré et al., 2003*). Consistently, LAMA3 expression in S1 cells, could be abrogated by p53 inhibition with α-pifithrin (*Figure 9c*, *Figure 9—figure supplement 1a*), whereas ectopic addition of LN5 or lrECM, elevated *LAMA3* transcription (*Figure 9d*, *Figure 9—figure supplement 1b*) in parallel to the activation of wild-type p53 (*Figure 7a*, *Figure 7—figure supplement 1a*).

To see whether there is a correlation between the wild-type p53 and LAMA3 levels *in vivo*, we performed immunohistochemical analyses of primary breast tissues using antibodies against the wildtype p53 (Clone pAb1620) and LAMA3 (Clone 546215). All normal breast tissue sections were stained strongly with both antibodies (*Figure 9—figure supplement 1c*). The reciprocity between LN5 and wild-type p53 remains strong even as cells progress to malignancy. The levels of the two proteins fell in parallel in the tumor samples (R = 0.51, p<0.0001, n = 117) (*Figure 9e*).

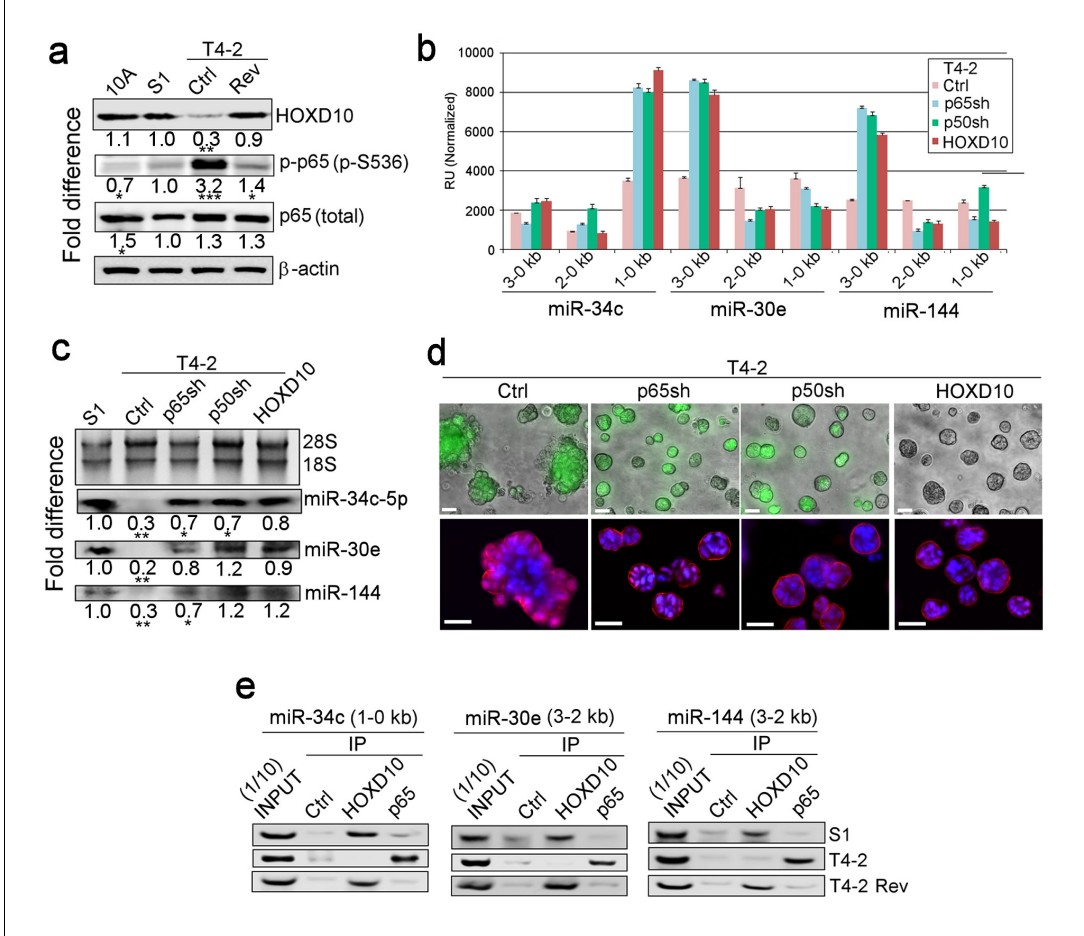

**Figure 4.** HOXD10 and NFκB regulate expression of the three miRNAs in opposite directions. (**a**) Representative western blot result (n = 3) for HOXD10 level, phosphorylation of p65 subunit of NFκB (p-p65, S536), and total p65 level in MCF10A, S1, T4-2 and T4-2 Rev (treated with AG1478) cells. Fold difference was determined with respect to S1 cells. *p<0.05 and ***p<0.001. Note the opposing patterns of HOXD10 and p-p65 levels. (**b**) Activities of different miRNA promoters (n = 3) in Ctrl, P65sh-, p50sh- or HOXD10-expressing T4-2 cells. Note that a 1–0 kb fragment of miR-34c promoter, a 3–0 kb fragment of miR-30e promoter and a 3–0 kb fragment of miR-144 promoter (the same regions activated in S1 and T4-2 Rev cells, ***Figure 3—figure supplement 2a***) were activated in p65sh-, p50sh- or HOXD10-expressing T4-2 cells. Data are represented as mean ± SEM. (**c**) Representative northern blot result (n = 3) for the expression of miR-34c-5p, miR-30e and miR-144 in Ctrl, p65sh-, p50sh- or HOXD10-expressing T4-2 cells. 28S and 18S serve as loading controls. Please note that for miR-30e and miR-144, both the sense (5 p) and antisense strands (3 p) are involved. *p<0.05 and **p<0.01. (**d**) Ctrl, p65sh-, p50sh- or HOXD10-expressing T4-2 cells grown in lrECM. (Top) Phase images overlaid with FITI to indicate transduced cells. (Bottom) red — integrin α6; blue — DAPI. Scale bars: 20 μm. See the quantification in ***Figure 4—figure supplement 1c***. (**e**) Representative result for ChIP analysis (n = 3) for the binding of HOXD10 and the NFκB p65 subunit on the miRNA promoters. Note that binding of HOXD10 and p65 to the mRNA promoters are mutually exclusive. See the quantification in ***Figure 4—figure supplement 1e***. For each analysis, replicate experiments (n = 3) were performed, and representative data are shown.

DOI: https://doi.org/10.7554/eLife.26148.015

The following figure supplement is available for figure 4:

**Figure supplement 1.** HOXD10 and NFκB positively and negatively regulate the miRNA expression, respectively.

DOI: https://doi.org/10.7554/eLife.26148.016

The essential and prominent steps of the acinar circuitry are shown in the schematic presented in ***Figure 10***.

## Discussion

The ability to phenotypically revert breast cancer cells by inhibiting a single signaling pathway in 3D lrECM has provided us with the means to identify additional major signaling pathways that must

**Table 7.** Decoy sequences of NFκB and HOXD10 for each miRNA promoter.

| miRNA promoter | TXN bound | Start from TSS | End from TSS | Predicted binding sequence | Wt decoy sequence (5′ → 3′) (Ds DNA) | Mt decoy sequence (5′→ 3′) (Ds DNA) | Transfected T4-2 cells |
|---|---|---|---|---|---|---|---|
| | Scramble | | | | F  TTGCCGTACCTGACTTAGCC | | |
| | | | | | R  GGCTAAGTCAGGTACGGCAA | | |
| miR-34c | NFκB | −787 | −778 | AGGGAATCAA | F  CCTTGAA<u>AGGGAATCAA</u>TCC | F  CCTTGAA<u>AtGtAcTaAc</u>TCC | Ctrl (pDCF1) |
| | | | | | R  GGA<u>TTGATTCCCT</u>TTCAAGG | R  GGA<u>gTtAgTaCa</u>TTTCAAGG | |
| | | −769 | −760 | TGGGAAGTTT | F  CCTTGAA<u>TGGGAAGTTT</u>TCC | F  CCTTGAA<u>TtGtAcGgTg</u>TCC | Ctrl (pCDF1) |
| | | | | | R  GGA<u>AAACTTCCCA</u>TTCAAGG | R  GGA<u>cAcCgTaCa</u>ATTCAAGG | |
| | | −427 | −418 | TGGGAACCTT | F  CCTTGAA<u>TGGGAACCTT</u>TCC | F  CCTTGAA<u>TtGtAcCaTg</u>TCC | Ctrl (pCDF1) |
| | | | | | R  GGA<u>AAAGGTTCCCA</u>TTCAAGG | R  GGA<u>cAtGgTaCa</u>ATTCAAGG | |
| | | −64 | −55 | TGGGAAGCCG | F  CCTTGAA<u>TGGGAAGCCG</u>TCC | F  CCTTGAA<u>TtGtAcGaCt</u>TCC | Ctrl (pCDF1) |
| | | | | | R  GGA<u>CGGCTTCCCA</u>TTCAAGG | R  GGAA<u>GtCgTaCa</u>ATTCAAGG | |
| | | −56 | −47 | CGCTTTCCCA | F  CCTTGAA<u>CGCTTTCCCA</u>TCC | F  CCTTGAA<u>CtCgTgCaCc</u>TCC | Ctrl (pCDF1) |
| | | | | | R  GGA<u>TGGGAAAGCG</u>TTCAAGG | R  GGA<u>gGtGcAcGa</u>GTTCAAGG | |
| | | -3 | 6 | GGGGAATGAG | F  CCTTGAA<u>GGGGAATGAG</u>TCC | F  CCTTGAA<u>GtGtAcTtAt</u>TCC | HOXD10/ pCDF1 |
| | | | | | R  GGA<u>CTCATTCCCC</u>TTCAAGG | R  GGA<u>aTaAgTaCa</u>CTTCAAGG | |
| | HOXD10 | −864 | −855 | AGTTTGTATT | F  CCTTGAA<u>AGTTTGTATT</u>TCC | F  CCTTGAA<u>AtTgTtTcTg</u>TCC | HOXD10/ pCDF1 |
| | | | | | R  GGA<u>AATACAAACT</u>TTCAAGG | R  GGA<u>cAgAaAcAa</u>TTTCAAGG | |
| | | −385 | −376 | CCCTTCTATT | F  CCTTGAA<u>CCCTTCTATT</u>TCC | F  CCTTGAA<u>CaCgTaTcTg</u>TCC | HOXD10/ pCDF1 |
| | | | | | R  GGA<u>AATAGAAGGG</u>TTCAAGG | R  GGA<u>cAgAtAcGt</u>GTTCAAGG | |
| miR-30e | NFκB | −2888 | −2879 | GATATTCCCA | F  CCTTGAA<u>GATATTCCCA</u>TCC | F  CCTTGAA<u>GcTcTgCaCc</u>TCC | Ctrl (pCDF1) |
| | | | | | R  GGA<u>TGGGAATATC</u>TTCAAGG | R  GGA<u>gGtGcAgAg</u>CTTCAAGG | |
| | HOXD10 | −2553 | −2544 | TGGTTGTATT | F  CCTTGAA<u>TGGTTGTATT</u>TCC | F  CCTTGAA<u>TtGgTtTcTg</u>TCC | HOXD10/ pCDF1 |
| | | | | | R  GGA<u>AATACAACCA</u>TTCAAGG | R  GGA<u>cAgAaAcCa</u>ATTCAAGG | |
| | | −2230 | −2221 | GCGTGATATT | F  CCTTGAA<u>GCGTGATATT</u>TCC | F  CCTTGAA<u>GaGgGcTcTg</u>TCC | HOXD10/ pCDF1 |
| | | | | | R  GGA<u>AATATCACGC</u>TTCAAGG | R  GGA<u>cAgAgCcCt</u>CTTCAAGG | |
| | | −2122 | −2113 | TTTTTTTATT | F  CCT TGAA<u>TTTTTTTATT</u>TCC | F  CCTTGAA<u>TgTgTgTcTg</u>TCC | HOXD10/ pCDF1 |
| | | | | | R  GGA<u>AATAAAAAAA</u>TTCAAGG | R  GGA<u>cAgAcAcAc</u>ATTCAAGG | |
| | | −1967 | −1967 | TACTCATATT | F  CCTTGAA<u>TACTCATATT</u>TCC | F  CCTTGAA<u>TcCgCcTcTg</u>TCC | HOXD10/ pCDF1 |
| | | | | | R  GGA<u>AATATGAGTA</u>TTCAAGG | R  GGA<u>cAgAgGcGg</u>ATTCAAGG | |
| miR-144 | NFκB | −3207 | −3198 | AGGGAATTTG | F  CCTTGAA<u>AGGGAATTTG</u>TCC | F  CCTTGAA<u>AtGtAcTgTt</u>TCC | Ctrl (pCDF1) |
| | | | | | R  GGA<u>CAAATTCCCT</u>TTCAAGG | R  GGA<u>aAcAgTaCa</u>TTTCAAGG | |
| | HOXD10 | −2850 | −2841 | AATAGAATGA | F  CCTTGAA<u>AATAGAATGA</u>TCC | F  CCTTGAA<u>AcTcGcAgGc</u>TCC | HOXD10/ pCDF1 |
| | | | | | R  GGA<u>TCATTCTATT</u>TTCAAGG | R  GGA<u>gCcTgCgAg</u>TTTCAAGG | |
| | | −2715 | −2706 | AATACAAAAA | F  CCTTGAA<u>AATACAAAAA</u>TCC | F  CCTTGAA<u>AcTcCcAcAc</u>TCC | HOXD10/ pCDF1 |
| | | | | | R  GGA<u>TTTTTGTATT</u>TTCAAGG | R  GGA<u>gTgTgGgAg</u>TTTCAAGG | |
| | | −2412 | −2403 | CCATATTATT | F  CCTTGAA<u>CCATATTATT</u>TCC | F  CCTTGAA<u>CaAgAgTcTg</u>TCC | HOXD10/ pCDF1 |
| | | | | | R  GGA<u>AATAATATGG</u>TTCAAGG | R  GGA<u>cAgAcTcTt</u>GTTCAAGG | |

*Table 7 continued on next page*

*Table 7 continued*

| miRNA promoter | TXN bound | Start from TSS | End from TSS | Predicted binding sequence | Wt decoy sequence (5′ → 3′) (Ds DNA) | Mt decoy sequence (5′→ 3′) (Ds DNA) | Transfected T4-2 cells |
|---|---|---|---|---|---|---|---|
| | | −2336 | −2327 | AATAAGAGTA | F CCTTGAA<u>AATAAGAGTA</u>TCC | F CCTTGAA<u>AcTcAtAtTc</u>TCC | HOXD10/pCDF1 |
| | | | | | R GGA<u>TACTCTTATT</u>TTCAAGG | R GGA<u>gAaTaTgAg</u>TTTCAAGG | |
| | | −2312 | −2303 | ATTTATTATT | F CCTTGAA<u>ATTTATTATT</u>TCC | F CCTTGAA<u>AgTgAgTcTg</u>TCC | HOXD10/pCDF1 |
| | | | | | R GGA<u>AATAATAAAT</u>TTCAAGG | R GGA<u>cAgAcTcAc</u>TTTCAAGG | |

Note: the transcription factor binding sites are underlined, whereas mutated nucleotides are indicated in lower case.

DOI: https://doi.org/10.7554/eLife.26148.018

integrate for the formation of 'phenotypically normal' human breast acini (*Weaver et al., 1997*; *Muschler et al., 2002*; *Beliveau et al., 2010*; *Bissell and Hines, 2011*; *Lee et al., 2012*; *Tanner et al., 2012*; *Becker-Weimann et al., 2013*). Here, we set out to develop a blueprint for how the breast cells interpret their interactions with the ECM proteins LN1 and LN5. The LNs trigger the signaling cascade leading to reciprocal communications between the ECM and TFs essential for mammary morphogenesis.

To do this, we used a unique breast cancer progression series, HMT3522: non-malignant S1, malignant T4-2 and T4-2 reverted to non-malignant phenotype (using five signaling inhibitors of oncogenic pathways, where addition of single inhibitors could revert the malignant phenotype). We observed that, although T4-2 Rev cells have similar phenotypes, their gene expression patterns were very different (*Becker-Weimann et al., 2013*). Nevertheless, a comparison of the gene arrays of the five T4-2 revertants identified a group of 60 similar genes that are also expressed in S1 cells (*Figure 1d*) (*Becker-Weimann et al., 2013*). This led us to propose that the common denominator of reversion had to contain a number of miRNAs that regulate this gene subset. We thus devised miRNA expression arrays and identified 10 miRNAs that fit the above category (*Figure 2a*). This result, together with the literature search (*Lu et al., 2005*) and our analysis of miRNA expression in normal- vs. cancerous- breast tissues (*Figure 2b*), identified three miRNAs (miR-34c-5p, −30e, and −144) that were shown to be severely downmodulated in primary breast tumors (*Figure 2b and c*, *Figure 2—figure supplement 1*) (*Lu et al., 2005*). As expected, restoration of any of these three miRNAs in T4-2 cells led to phenotypic reversion in lrECM (*Figure 2e*). We utilized these miRNAs as the focal starting point to dissect the fundamental reciprocal pathways necessary for the formation and maintenance of breast tissue architecture.

It has been long known that diverse biological activities in development are regulated by tissue–tissue and tissue–microenvironment interactions and signaling (*Wessells, 1977*; *Chiquet-Ehrismann et al., 1986*; *Howlett and Bissell, 1993*; *Hogan, 1999*; *Bhat and Bissell, 2014*). During development, different cell types communicate and coordinate with each other through negative and positive feedback regulations. Within a given tissue, there are also negative and positive operators that must be regulated constantly to maintain homeostasis and quiescence as we demonstrated here. In addition, similar to movements that are being discovered in the formation of embryos during development (*Haigo and Bilder, 2011*), tissue formation starts with cells moving within a soft microenvironment such as lrECM, as we and others observed for mammary acini; we termed this 'coherent angular motion' (*Tanner et al., 2012*). CAMo creates polarity and adhesion by interacting with exogenous ECM to lay down its own endogenous tissue-specific ECM (*Tanner et al., 2012*).

The balance and integration of the different signaling pathways and dynamic interactions between epithelial cells and the ECM drive the remodeling of the ECM, including formation of the BM that helps to anchor the epithelia (*Weaver et al., 2002*) and that protects the cells within the tissues from apoptosis. Such changes in the ECM regulate cell proliferation, survival, migration, shape and adhesion, ultimately sculpting and maintaining tissue architecture (*Wessells, 1977*; *Chiquet-Ehrismann et al., 1986*; *Howlett and Bissell, 1993*; *Hogan, 1999*; *Bhat and Bissell, 2014*; *Haigo and Bilder, 2011*; *Weaver et al., 2002*; *Daley and Yamada, 2013*). There are important

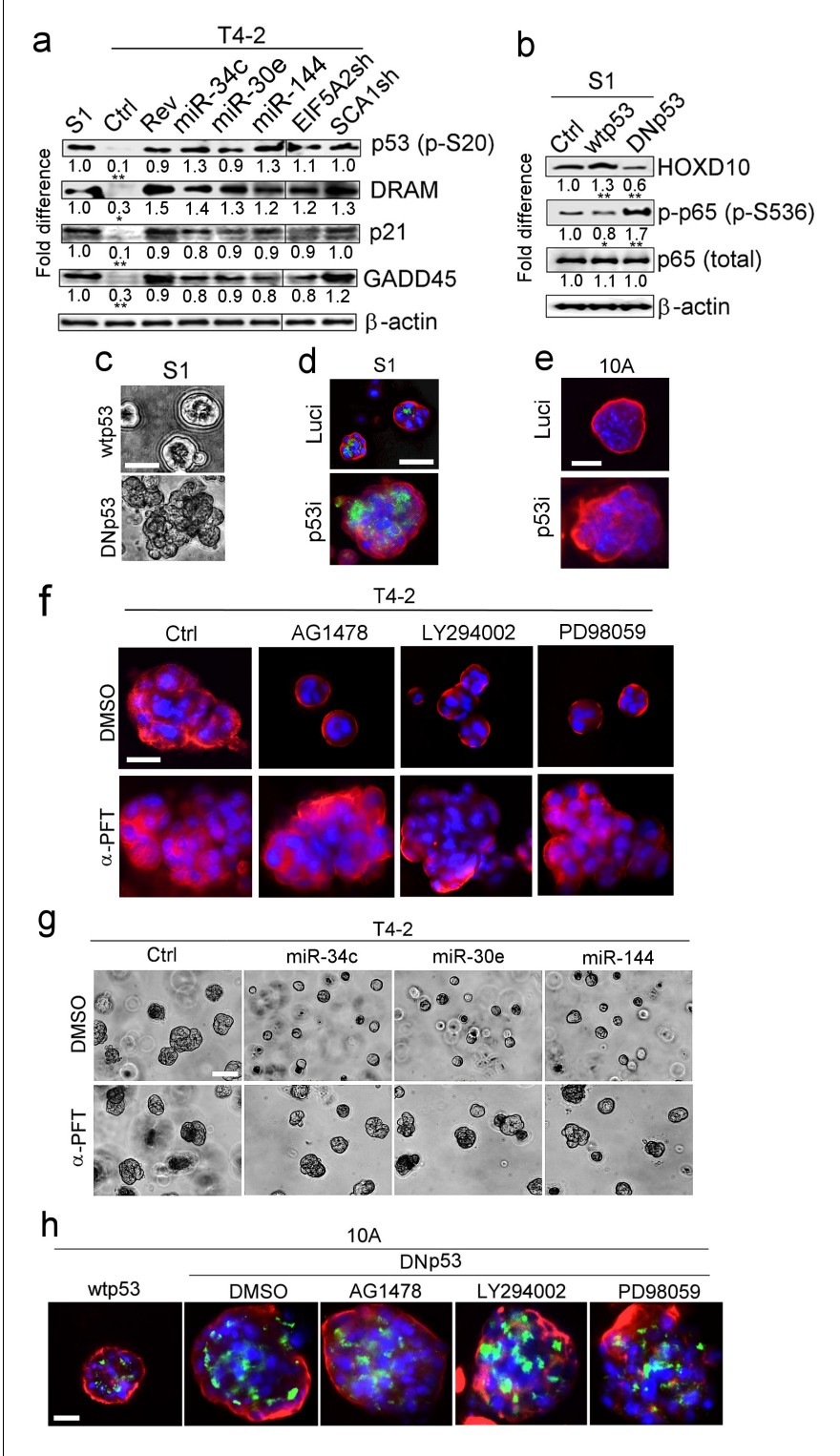

**Figure 5.** p53 activation is another requirement for tumor-cell reversion by the miRNAs. (a) Representative result of western analysis (n = 3) for the level and activation of proteins in the p53 pathway [p53 (p-Ser20), DRAM, p21 and GADD45] in S1, T4-2 and T4-2 Rev cells. T4-2 Rev cells include those treated with AG1478, those expressing the three miRNAs and those depleted of the two target genes (EIF5A2 and SCA1). Fold difference was determined with respect to S1 cells. *p<0.05 and **p<0.01. (b) Representative result of western analysis (n = 3) for the activities of HOXD10 and p65 (p-p65) in Ctrl S1 cells and in S1 cells that were overexpressing the wild-type (wt) or dominant-negative mutant (DN) p53. Note the opposing effects of wtp53 or DNp53 expression on HOXD10 vs.

*Figure 5 continued on next page*

*Figure 5 continued*

p-p65. Fold difference was determined with respect to Ctrl S1. *p<0.05 and **p<0.01. (c) Representative image for S1 cells overexpressing wtp53 (top) or DNp53 (bottom) grown in lrECM. Ctrl S1 cells were un-transduced. See the quantification of 3D colony size in *Figure 5—figure supplement 1a*. (d) Representative image for Ctrl S1 (top) and S1 cells depleted of p53 (bottom) grown in lrECM. Red — α6, green; apical marker GM130. See the confirmation of p53 depletion and quantification of the 3D colony size in *Figure 5—figure supplement 1b*. (e) Representative image for Ctrl MCF10A cells (top) and MCF10A cells depleted of p53 (bottom) grown in lrECM. See the confirmation of p53 depletion and quantification of the 3D colony size in *Figure 5—figure supplement 1c*. (f) Representative image of 3D morphologies of T4-2 cells co-treated with a reverting agent [AG1478 (EGFR inhibitor), LY294002 (PI3K inhibitor) or PD98059 (MEK inhibitor)] along with p53 inhibitor α-pifithrin (α-PFT). Note that α-PFT treatment abrogated reversion of T4-2 cells. See the quantification of 3D colony size in *Figure 5—figure supplement 1d*. (g) Representative image of 3D morphologies of miRNA-overexpressing T4-2 cells co-treated with a reverting agent [AG1478 (EGFR inhibitor), LY294002 (PI3K inhibitor) or PD98059 (MEK inhibitor)] along with p53 inhibitor α-pifithrin (α-PFT). Note that α-PFT treatment abrogated the reversion of T4-2 cells after expressing miRNAs. See the quantification of the 3D colony size in *Figure 5—figure supplement 1e*. (h) Representative image of 3D morphologies of DNp53-expressing MCF10A cells treated with a reverting agent [AG1478 (EGFR inhibitor), LY294002 (PI3K inhibitor) or PD98059 (MEK inhibitor)]. Note that DNp53-expressing cells are more proliferative, fail to form acini and are resistant to a reverting agent. See the quantification of the 3D colony size in *Figure 5—figure supplement 1f*. Red — integrin α6; green — Golgi marker, GM130; and blue — DAPI. Scale bars: 20 μm. For each analysis, replicate experiments (n = 3) were performed, and representative data are shown.

DOI: https://doi.org/10.7554/eLife.26148.019

The following figure supplement is available for figure 5:

**Figure supplement 1.** p53 activation is essential for acinar formation and tumor-cell reversion.
DOI: https://doi.org/10.7554/eLife.26148.020

---

differences, however, between developmental processes and tissue maintenance and renewal (*Howlett and Bissell, 1993*; *Hogan, 1999*; *Bhat and Bissell, 2014*; *Daley and Yamada, 2013*). Unlike the signaling pathways in development, the stability of the differentiated state does not appear to be hierarchical. Instead, it reflects the balance between growth and differentiation, between the negative and positive signaling pathways, and between the formation of a BM and the destruction of ECM by degrading enzymes that determines the stability of the differentiated state in the tissues.

Another novel finding here is that NO is a pivotal player in reciprocal cell–ECM interactions in breast morphogenesis, but tumor cells produce only a small amount of NO unless the architecture is re-established and the cells have reverted to a 'dormant state' (*Figure 7c–e and g*). This is a mimicry of differentiation-dependent tissue architecture. These findings demonstrate that NO production is a mechanistic link between proper architecture and proper function in breast tissues. Please see also the accompanying paper of *Ricca et al., 2018*, which describes how the reversion of T4-2 cells induced by a short period of compression in laminin is also mediated by NO production.

There are a few papers in the literature on connections between LN1 and NO in other tissues (*Rialas et al., 2000*; *Gloe et al., 1999*), and there are other reports of activation of p53 by high levels of exogenous NO (*Forrester et al., 1996*; *Gordon et al., 2001*; *Wang et al., 2003*). To our knowledge, however, there are no reports of endogenous NO as a critical link in the formation of mammary epithelium and its role in stability of the tissue architecture.

It is crucial to note that the levels of NO produced endogenously in response to LNs in our studies, as well as the exogenous NO levels required for the reciprocal loop we describe here, are at least *500-fold lower* than those used in the literature (*Forrester et al., 1996*; *Gloe et al., 1999*; *Gordon et al., 2001*). As stated long ago, differences in quantity of such magnitudes becomes a change in quality and hence have appreciable consequence (*Bissell, 1981*).

NO has been reported to play an important role during lactation. Increased levels of NO are produced by the mammary gland of postpartum mammals (*Akçay et al., 2002*). NO promotes blood flow and the nutrient uptake of mammary glands for milk production (*Kim and Wu, 2009*). NO is also proposed to facilitate milk ejection by inducing contraction of myoepithelial cells (MEPs) in mammary glands as well as smooth muscle cells in the stroma (*Iizuka et al., 1998*; *Adriance et al., 2005*; *Tezer et al., 2012*). In addition, NO is secreted into the breast milk as an essential component

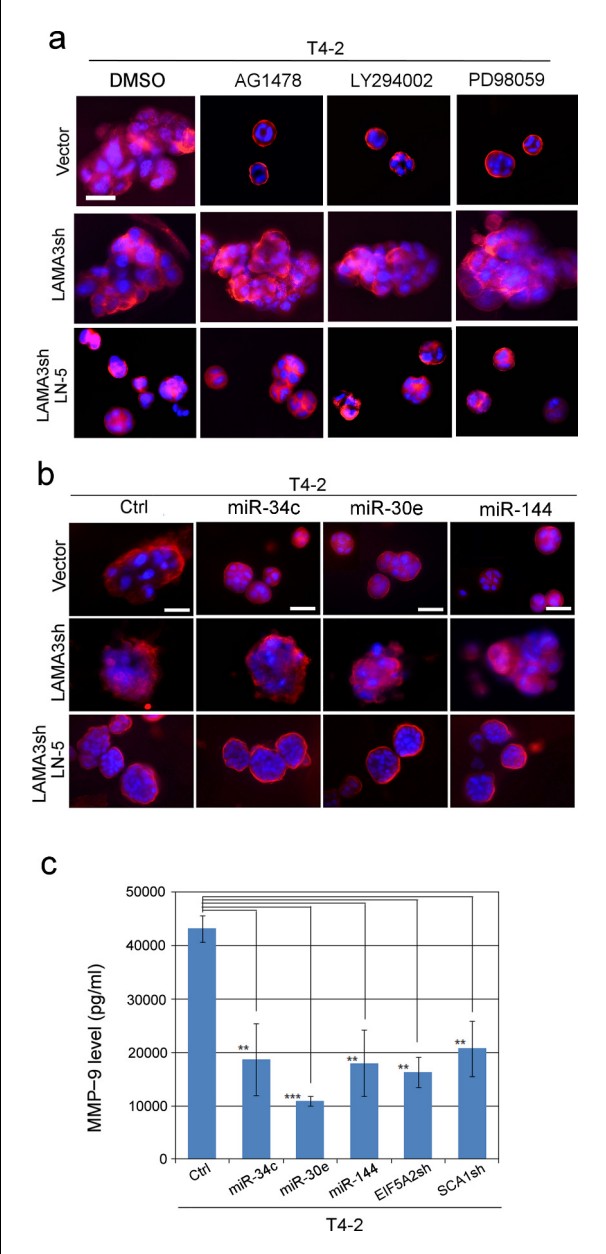

**Figure 6.** MMP-9 degrades LN5 and prevents tumor-cell reversion; miRNAs' ability to revert T4-2 cells is connected to inhibition of MMP-9 and the rescue of LN5. (a) Representative images of repricate experiments (n = 3) of 3D morphologies of T4-2 cells (vector, LAMA3sh, LAMA3sh + LN5) reverted with a reverting agent, AG1478, LY294002 or PD98059. See the quantification of the 3D colony size in *Figure 6—figure supplement 2a*. Note that the depletion of LAMA3 abrogated the reverting effect, which was rescued by ectopic addition of LN5 (1 μg/ml). Red — α6 integrin; blue — DAPI. Scale bars: 20 μm. (b) Representative images of replicate experiments (n = 3) of 3D morphologies of T4-2 cells (vector, LAMA3sh, LAMA3sh + LN5) overexpressing individual miRNAs. See the quantification of the 3D colony size in *Figure 6—figure supplement 2b*. Note that the depletion of LAMA3 abrogated the reverting effect of the miRNAs, which was rescued by ectopic addition of LN5. (c) The mean MMP-9 level (n = 9) in the CM of T4-2 cells, T4-2 cells expressing one of the three miRNAs or T4-2 cells depleted of the two target genes, EFI5A2 and SCA1, was determined 24 hr after addition of lrECM (5% Matrigel). The concentration of MMP-9 was determined using the MMP-9 standard. Data represented as mean ± SEM. **p<0.01 and ***p<0.001.

DOI: https://doi.org/10.7554/eLife.26148.021

The following figure supplements are available for figure 6:

*Figure 6 continued on next page*

*Figure 6 continued*

**Figure supplement 1.** LN5 expression is upregulated in T4-2 cells overexpressing the three miRNAs or depleted of their target genes.

DOI: https://doi.org/10.7554/eLife.26148.022

**Figure supplement 2.** LN5 expression is required for reversion of tumor cells in 3D.

DOI: https://doi.org/10.7554/eLife.26148.023

for immunity in neonatal growth (*Hord et al., 2011*). Using 3D cultures and *ex vivo* cultures of human mammary glands, we showed here that NO also plays additional and significant roles in breast morphogenesis (*Figure 8a, b and d*).

Importantly, NO production was specific to LNs and was not induced by collagen (*Figure 7c-e*). We and others had shown previously that LNs and COL1 elicit opposite actions on epithelia (*Gudjonsson et al., 2002*; *Oktay et al., 2000*; *Chamoux et al., 2002*). We showed here that LNs activate NOS-1 (*Figure 7a*, *Figure 7—figure supplement 1a*), supporting previous observations by others that NOS-1 is expressed in the mammary tissue at appreciable levels — in particular in MEPs during pregnancy and lactation in humans (*Tezer et al., 2012*) and rodents (*Iizuka et al., 1998*; *Islam et al., 2009*; *Wockel et al., 2005*). As the molecule that appears to be responsible for linking LNs to NOS-1, we speculate the involvement of the LN receptor, dystroglycan (DG), which is known to form a multi-protein complex involving LNs and NOS-1, in mediating the mechanotransduction of muscle cells (*Rando, 2001*; *Garbincius and Michele, 2015*). We had shown previously that DG also plays a critical signaling role in breast epithelial cells (*Muschler et al., 2002*). DG anchors the BM protein, in particular LNs, to the cell surface, allowing for LN polymerization and transduction of signals for the formation of polarized colonies (*Weir et al., 2006*). Such DG–LN interaction is impaired in different types of cancer cells and correlates with poorer patient prognosis (*Akhavan et al., 2012*; *Esser et al., 2013*).

Form and function are maintained in adult organs throughout most of the life of the organism, despite constant mutations and damage from environmental assaults and aging. To maintain the correct tissue function throughout the lifetime of the organisms, signaling pathways have to integrate in order to prevent chaos and malfunction. Evolution has packed much wisdom and specificity onto the ECM, which appears to instruct the chromatin to change shape and thus also gene expression, as seen in *Figure 4e*. When cells on flat surfaces receive LNs, not only their shape, but also many of their signaling pathways are altered (*Figure 1*) (*Bissell et al., 2005*); growth must stop in many tissues (*Spencer et al., 2011*; *Fiore et al., 2017*) and differentiation and cell death must be coordinated. It is now clear that narratives that are based solely on linear and irreversible regulatory dynamics cannot satisfactorily explain the reality *in vivo* (*Hogan, 1999*). It is also clear that, at the last analysis, it is the 3D architecture of the tissue itself that is the message (*Hagios et al., 1998*).

## Materials and methods

### Cell lines

Cell lines of the HMT3522 breast cancer progression series (S1 and T4-2) were provided by O.W. Petersen (Laboratory of Tumor Endocrinology, The Fibiger Institute, Copenhagen, Denmark) (*Briand et al., 1996*). The cell lines were authenticated by genome sequencing by the provider. Mycoplasma testing was negative. MCF10A cells were obtained from the Karmanos Cancer Institute (Detroit, MI, USA) under a Material Transfer Agreement. The cell lines was authenticated by the provider. Mycoplasma testing was negative.

### Cell culture and reagents

The isogenic cell lines of the HMT3522 human breast cancer progression series, non-malignant S1 and malignant T4-2 cells, were maintained as described previously (*Briand et al., 1996*). This cell line series was established in an attempt to recapitulate the stochastic and prolonged nature of breast cancer progression by continuously culturing S1 cells, derived from reduction mammoplasty, in the absence of serum, followed by EGF removal and injection into mice, to give rise to T4-2 cells (*Briand et al., 1996*). For 3D culture experiments, S1 and T4-2 cells were seeded at the density of

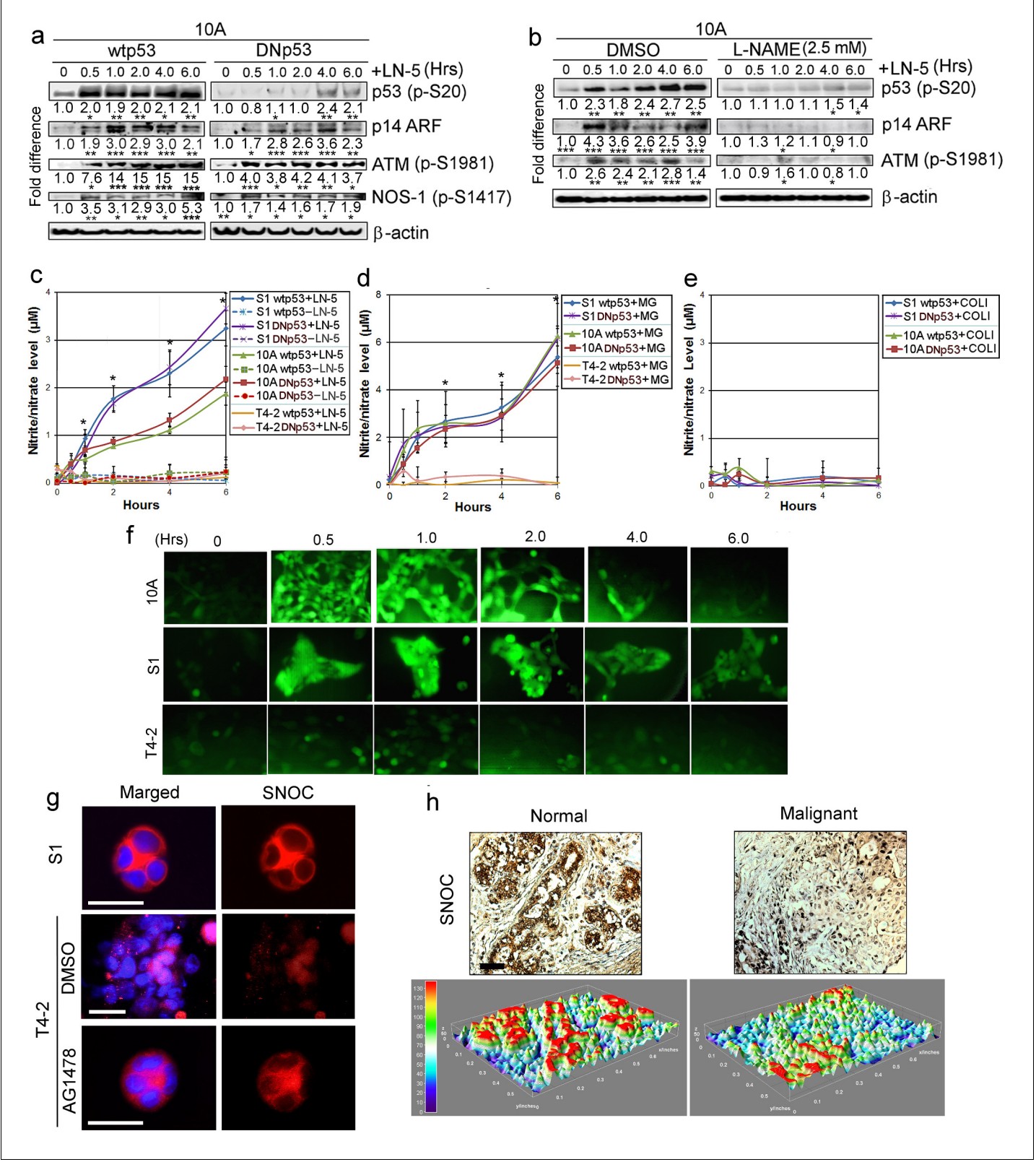

**Figure 7.** LNs activate p53 through production of NO. (a) Representative results of western analysis (n = 3) for MCF10A cells expressing either wtp53 or DNmtp53, showing the level of activated p53 (p-Ser20) after addition of exogenous LN5 (1 μg/ml). Other lanes indicate the levels of p14 ARF and activation of ATM and NOS-1 after LN5 addition. Fold differences were determined first by normalization with respect to β-actin and then by normalization with respect to the value at time 0. *p<0.05; **p<0.01 and ***p<0.001. See similar results after the addition of Matrigel (5%) in *Figure 7—*

*Figure 7 continued on next page*

*Figure 7 continued*

*figure supplement 1a*. (**b**) Representative results of western analysis (n = 3) showing the levels of activation of ATM, p14 ARF and p53 in the absence or the presence of an NO inhibitor, L-NAME. (**c–e**) The mean level of NO metabolites (nitrite or nitrate, n = 9) in CM after addition of LN5 (1 µg/ml) (**c**), lrECM (5% Matrigel) (**d**) or COL1 (500 µg/ml) (**e**); please note that this amount is equivalent to the total protein level of 5% Matrigel. Nitrite or nitrate level was determined with a fluorescent probe DAN using the nitrite/nitrate standard. Data are represented as mean ± SEM. *p<0.05. Note that S1 and MCF10A cells, but not T4-2 cells, produced NO in response to LN5 (**c**) and lrECM (**d**), irrespective of whether they express wild-type or mutant p53. (**e**) COL1 did not induce NO production. (**f**) Representative images (n = 3) showing the level of intracellular NO in MCF10A, S1 and T4-2 cells after addition of 5% MG, as determined with a fluorescent NO probe DAF-FM DA. See the quantification of NO level in *Figure 7—figure supplement 1b*. (**g**) Representative images (n = 3) of S-nitrosocysteine (SNOC, NO indicator)-stained HMT3522 cells (S1, T4-2 and T4-2 Rev cells with AG1478) in lrECM cultures for 1 wk. Note that SNOC was enriched in the basolateral surface of S1 and T4-2 Rev cells, whereas it was weakly diffused in T4-2 cells treated with DMSO. Red —SNOC; blue — nuclei. Scale bars: 20 µm. (**h**) Representative images of normal (n = 8) vs. cancerous (n = 32) breast tissues stained for SNOC presented as IHC (top) and heat map of surface plot (bottom). Positive staining (intensity >+1): 8/8 for normal vs. 8/32 for cancerous tissues. Scale bars: 50 µm.

DOI: https://doi.org/10.7554/eLife.26148.024

The following figure supplement is available for figure 7:

**Figure supplement 1.** Acinar-forming breast cells produce NO in response to LNs.

DOI: https://doi.org/10.7554/eLife.26148.025

---

$2.5 \times 10^4$ cells/cm$^2$ and $1.8 \times 10^4$ cells/cm$^2$, respectively, in growth factor reduced Matrigel (Corning, NY, USA) and maintained for 10 days with the addition of fresh medium on alternate days. For T4-2 reversion, EGFR inhibitor AG1478 (EMD Millipore, Burlington, MA, USA) was used at 350 nM, PI3K inhibitor LY294002 at 8 µM, and MEK inhibitor PD98059 at 20 µM (*Lee et al., 2012*). For p53 inhibition, 30 µM α-PFT (α-pifithrin, Sigama-Aldrich, St. Louis, MO, USA) was used. For inhibition of NO production, cells were treated with 2.5 mM L-NAME (N$_\omega$-Nitro-L-arginine methyl ester hydrochloride, Sigma- Aldrich); for induction of NO production, 10 µM SNAP (S-Nitroso-N-acetyl-DL-penicillamine, Sigma- Aldrich) was used.

## miRNA array

miRNA expression profiling was performed using the RT$^2$miRNA PCR Array System (Qiagen, Inc. USA, Germantown, MD, USA) on the MyiQ Single-Color Real-Time PCR platform (Bio-Rad, Hercules, CA, USA). Briefly, $1.0 \times 10^6$ cells were grown in 1.2 ml Matrigel in 30 mm-plates for 10 days (for T4-2 Rev, 350 nM AG1478 was added). The medium was removed and cells were scraped off from the dish with 2 ml phosphate-buffered saline (PBS) with 5 mM EDTA. Cells were spun down to harvest pellets, which were repetitively washed with ice-cold PBS + EDTA until the Matrigel was dissolved. The total RNA was extracted with 1 ml Trizol (Life Technologies) and purified with an RNeasy plus mini kit (Qiagen, Inc, USA) according to the manufacturers' protocols. cDNA was generated from 4 µg of RNA using the RT$^2$miRNA First Strand Kit (SABiosciences), mixed with SYBR Green Master Mix (SABioseiciences) and loaded onto an array with 98 wells. Real-time PCR was performed according to the manufacturer's instructions, and data analysis was performed using the manufacturer's PCR Array Data Analysis Web Portal (Qiagen, Inc, USA).

## Northern analysis

Northern analysis of miRNAs was performed using the DIG detection system from Roche. Briefly, $1.0 \times 10^6$ cells/30-mm plate were grown in 1.2 ml Matrigel in triplicates for 10 days (for T4-2 Rev, 350 nM AG1478 was added). Cells were scraped off from the dish with PBS with 5 mM EDTA, spun down and washed with PBS + EDTA until the Matrigel was dissolved. The total RNA was extracted with 1 ml Trizol (ThermoFisher Scientific, Waltham, MA, USA). 20 µg of RNA was separated by denaturing polyacrylamide TBE-Urea gel electrophoresis (ThermoFisher Scientific) and electroblotted onto Bright-Star nylon membrane (Ambion) with 0.5% TBE for 2 hr. The membrane was rinsed in 2xSSC buffer, UV cross-linked at 120 mJ/cm$^2$, dried and stored between filter papers. LNA-modified DNA oligonucleotides complementary to the mature miRNA sequences (*Table 9*) were obtained from IDT and DIG-labeled using the DIG Oligonucleotide Tailing Kit (Roche Diagnostics, USA, Indianapolis, IN, USA). Using DIG Easy Hyb (Roche Diagnostics, USA), the membrane was prehybridized and hybridized with DIG-labeled probe at room temperature overnight. The membrane was washed

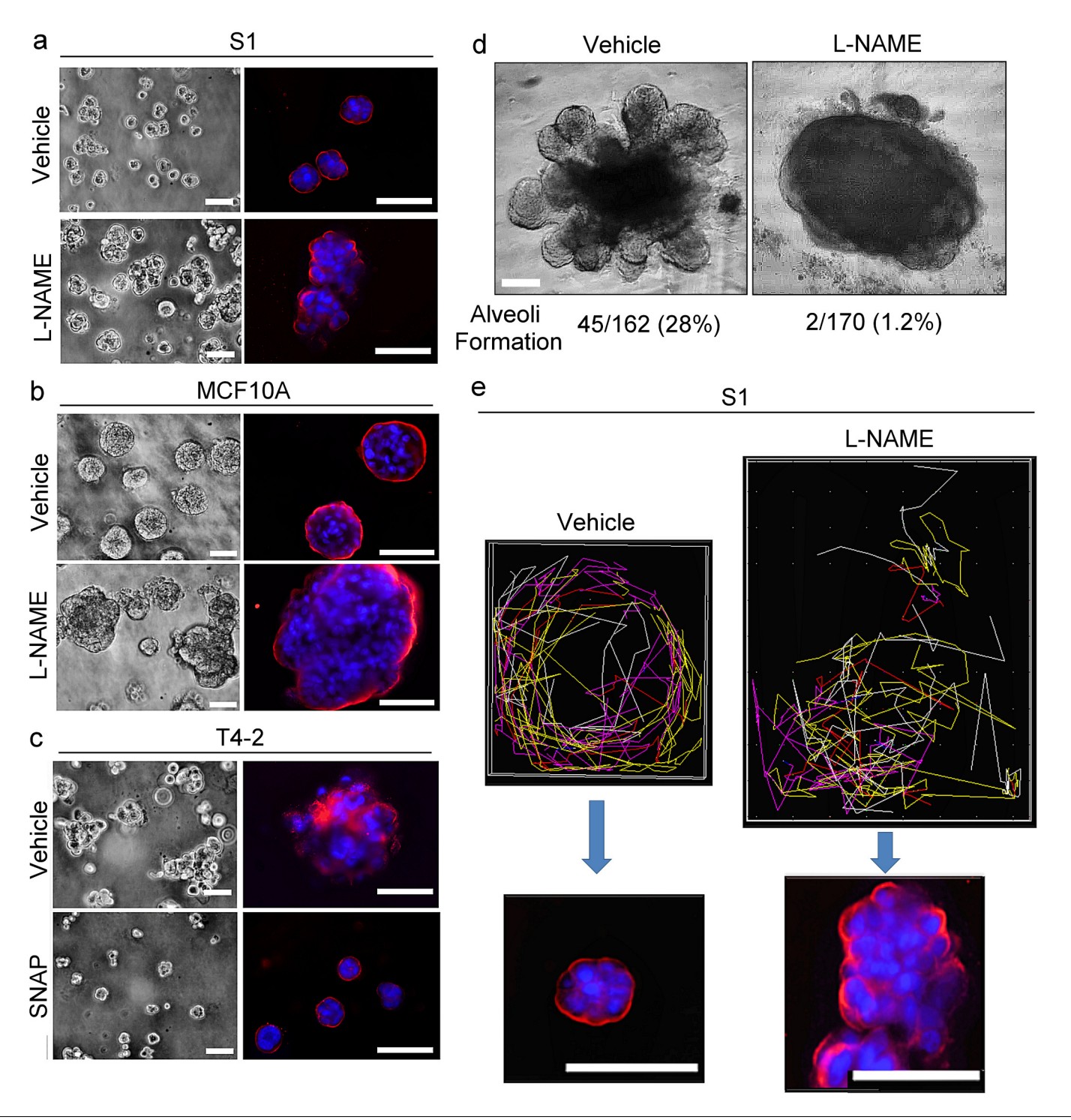

**Figure 8.** NO is involved critically in breast acinar formation and mammary gland morphogenesis. (**a–b**) Representative images (n = 3) of S1 (**a**) and MCF10A (**b**) cells grown in lrECM in the absence (vehicle only) or presence of an NO inhibitor, L-NAME (2.5 mM). Scale bars: 20 µm. (**c**) Representative image (n = 3) of T4-2 cells grown in lrECM in the absence (vehicle only) or presence of an NO donor, SNAP (10 µM). (**a–c**) (Left) Phase images. (Right) Cells stained for integrin α6 (red) and nuclei (blue). Scale bars: 20 µm. See the quantification of colony size in *Figure 8—figure supplement 1a–c*. (**d**) Representative images of *ex vivo* cultures of normal human mammary gland organoids grown in lrECM for 1 week. The numbers of organoids that underwent alveologenesis were 45 out of 162 vehicle-treated, and only 2 out of 170 L-NAME-treated organoids. (**e**) (Top) Representative result (n = 5) of tracking analysis for the movement of S1 cells during 48 hr of growth in lrECM in the absence (vehicle only) or presence of L-NAME. (Bottom) Representative image of the colonies (n = 5) formed by cells after respective treatments for 10 days. Red — integrin α6; blue —nucleus. Scale bars: 50

*Figure 8 continued on next page*

*Figure 8 continued*

µm. Note that vehicle-treated S1 cells moved in a coherent rotatory fashion in a confined area, whereas L-NAME-treated S1 cells moved in a disorganized fashion in a larger area.

DOI: https://doi.org/10.7554/eLife.26148.026

The following figure supplement is available for figure 8:

**Figure supplement 1.** NO is critical for breast cells to form growth-arrested colonies in 3D.

DOI: https://doi.org/10.7554/eLife.26148.027

in 10xSSC + 0.1% SDS four times and processed for DIG detection using the DIG Luminescent Detection Kit (Roche) according to the manufacturer's protocol.

## miRNA *in situ* hybridization (ISH)

miRNA *in situ* hybridization (ISH) was performed using the miRCURY LNA miRNA ISH Optimization kit for formalin-fixed paraffin embedded (FFPE) tissues (Qiagen, Inc, USA) and double-DIG-labeled detection probes for miR-34c-5p, miR-30e and miR-144 (EXIQON) on breast cancer tissue arrays containing paraffin-embedded sections of normal and malignant (stages II and III) tissues (US Bio-max, Inc, Rockville, MD, USA). Briefly, the tissue slides were heated at 60°C for 1 hr, deparaffinized in xylene and hydrated in alcohol series (100% to 70%). Slides were deproteinated with proteinase K for 20 min, fixed in 4% paraformaldehyde for 10 min and washed with 0.2% glycine in PBS for 5 min. Then, slides were incubated in imidazole buffer (0.13 M 1-methylimidazole, 300 mM NaCl, pH 8.0) for 10 min twice, in EDC solution (0.16M 1-ethyl-3-[3-dimethylaminopropyl] carbodiimide [EDC], pH 8.0) for 1.5 hr and washed with 0.2% glycine in PBS. Then, slides were dehydrated in an alcohol series (70% to 100%), hybridized with heat-denatured probes at room temperature overnight, and processed for DIG detection according to the manufacturer's protocol (Qiagen, Inc, USA). The slides were counterstained with Nuclear Fast Red and mounted with permount. Photomicrographs were taken with the Zeiss Axioskop Imaging Platform and Axion Vision software (Version 4.7).

## miRNA expression constructs

Lentivector-based precursor constructs for miR-34c-5p, miR-30e and miR-144 co-expressing copGFP were obtained from System Biosciences Palo Alto, CA, USA, and the virus particles to express each miRNA were produced according to the manufacturer's guideline.

## Gene-overexpressing lentiviral constructs

For construction of HOXD10-overexpressing lentivirus, the full-length human HOXD10 cDNA clone was obtained from Open Biosystems (Lafayette, CO, USA, Clone ID: 7262455). For construction of p53-overexpressing lentivirus, both wild-type and dominant-negative (A135V) p53 expression plasmids were obtained from Clontech. The coding region was PCR-amplified using the respective primers (*Table 9*). The PCR product was ligated into the *AscI/EcoR*1 site (for HOXD10) or the *BamH*I/*EcoR*1 site (for p53) of the PCDF1-MCS2-EF1-puro lentiviral vector (System Biosciences). For the SCA1- and EIF5A2-overexpression lentivirus construct, the cDNA clones were obtained from Origene (Rockville, MD, USA, Cat#RC222862 and Cat#RC206249, respectively) and cloned into PCDF1-MCS2-EF1-puro lentiviral vector at the *BamH*I/*EcoR*1 site using the Gibson assembly system and a DNA assembly kit (Cat# E5520S, NEB) with the primers designed on the NEB Builder Assembly Tool website as shown in *Table 9*.

## Gene knockdown by shRNA

For shRNA production, a double-stranded DNA oligonucleotide was generated from the respective sequences (*Table 9*). Sense and antisense oligonucleotides were annealed and ligated into *Bam*H1/*Eco*R1 site of pGreen puro lentival vector which co-expresses copGFP (System Biosciences).

## Lentivirus production and transduction

Lentivirus production and transduction of target cells were conducted following the guideline by System Biosciences. Briefly, lentivirus vector and packaging plasmid mix (System Biosciences) were transfected into 293FT cells (ThermoFisher Scientific) using Lipofectamine 2000. After 48 hr, medium

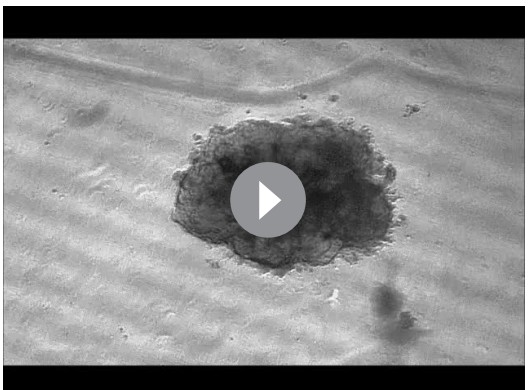

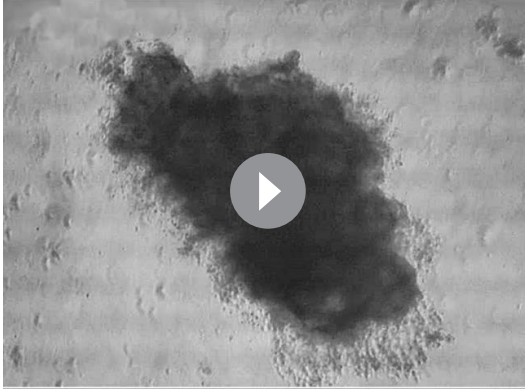

**Video 1.** Representative movie of alveologenesis of normal human mammary gland organoids in *ex vivo* 3D culture for 2 weeks.
DOI: https://doi.org/10.7554/eLife.26148.028

**Video 2.** Representative movie of impaired alveologenesis of L-NAME-treated normal human mammary gland organoids in *ex vivo* 3D culture for 2 weeks.
DOI: https://doi.org/10.7554/eLife.26148.029

was harvested, filtered and used to infect target cells with the addition of polybrene (10 µg/ml). The medium was replaced after 24 hr. At 72 hr post-infection, puromycin (0.5 µg/ml) was added for selection and maintained throughout the culturing period.

## RT-PCR

One million cells were grown in 1.2 ml Matrigel on a 30-mm plate for 10 days (for T4-2 Rev, 350 nM AG1478 was added). The medium was removed and cells were scraped off from the dish with 2 ml PBS with 5 mM EDTA. They were then spun down to harvest the cell pellet and repeatedly washed with PBS + EDTA until Matrigel was dissolved. The total RNA was extracted with 1 ml Trizol (ThermoFisher Scientific). cDNA was synthesized from 2 µg RNA using the SuperScript Double-Stranded cDNA Synthesis Kit (Invitrogen) and served as a template for PCR amplification with the respective primers (*Table 9*).

## Immunofluorescence staining

Immunofluorescence was performed as described previously (*Weaver et al., 1997*). Samples were incubated with primary antibody for 2 hr at room temperature in a humidified chamber. After intensive washing (three times, 15 min each) in 0.1% BSA, 0.2% Triton-X 100, 0.05% Tween 20, 0.05% NaN3 in PBS, fluorescence-conjugated secondary antibodies (Molecular Probes) were added for 1 hr at room temperature. Nuclei were stained with 0.5 ng/ml DAPI.

## Soft agar assay

One percent agar was mixed with the equivalent volume of 2x DMEM/F12 medium supplemented with all the additives necessary for culturing T4-2 cells (*Briand et al., 1996*) plus 20% FBS and 2% penicillin or streptomycin. 1 ml of the agar solution was poured into a 35 mm plate in triplicate and solidified. 0.7% agar solution equilibrated to 40°C was mixed with 2x growth medium and breast cancer cells at 7000 cells/ml and poured onto the base agar at 1 ml/plate. The solidified agar was covered with 500 µl growth medium and maintained in a 37°C humidified incubator for 14 d. The plates were stained with 0.01% crystal violet for 30 min, and colonies were counted under a dissecting microscope.

## Luciferase reporter assay

For generation of miRNA reporter constructs, the promoter regions of miRNA genes were obtained by PCR-amplifying BAC genomic clones [miR-34c (Ch11), PR11-794P6; miR-30e (Ch1), RP11-576N9; miR-144 (Ch17), RP11-832J20] using the respective primers (*Table 9*) and inserted into the pGL3 luciferase expression vector. Cells seeded at $5 \times 10^5$ cells/60-mm plate were transfected with 7 µg

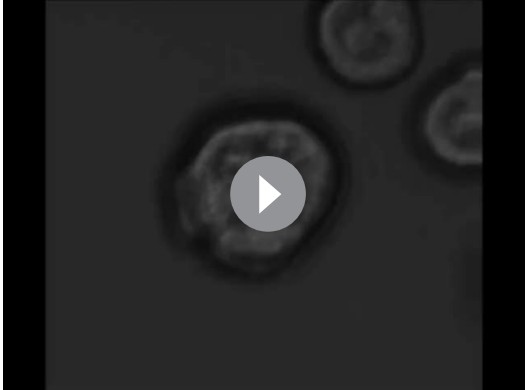 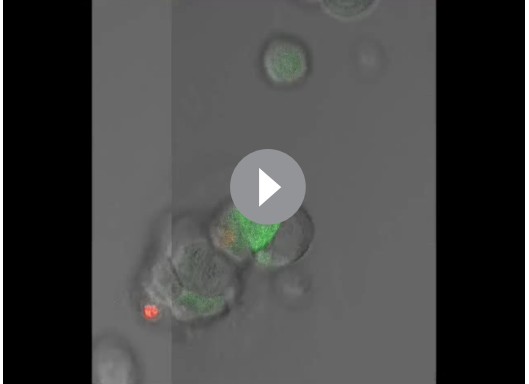

**Video 3.** Representative time-lapse movie of S1 cells undergoing coherent axial rotation in lrECM culture for 48 hr.
DOI: https://doi.org/10.7554/eLife.26148.030

**Video 4.** Representative time-lapse movie of L-NAME-treated S1 cells undergoing stochastic amoeboid movement in lrECM culture for 48 hr.
DOI: https://doi.org/10.7554/eLife.26148.031

of luciferase reporter and 0.5 µg of β-galactosidase plasmids using Xfect transfection reagent according to the manufacturer's protocol (Clontech, Mountain View, CA, USA). After 24 hr post transfection, the medium was replaced with the fresh medium containing 5% Matrigel and cells were maintained for another 24 hr (for T4-2 Rev, 350 nM AG1478 was added). Luciferase and β-galactosidase reporter activities were measured using a reporter assay kit (Promega, Madison, WI, USA).

## Decoy analysis

Wild-type miRNA decoy sequences (*Table 7*) were derived from the binding sites of NFκB or HOXD10 within the miRNA promoters predicted by AlGGEN PROMO software (see below). The sequence-specific binding of the two TFs was tested using mutant decoys (*Table 7*) that had point mutations in their core binding sequences. The forward and reverse oligonucleotides of decoys at 100 µM each were annealed in Duplex buffer (Integrated DNA Technologies, Coralville, IA, USA, CAT#11-05-01-12), and the same group of decoys was pooled. T4-2 cells were plated at $0.5 \times 10^5$/ 12 wells the day before transfection. NFκB decoys (scramble, WT or MT), along with miRNA promoters fused to luciferase (see above), were transfected into control T4-2 cells that had a high endogenous level of NFκB. HOXD10 decoys (scramble, WT or MT), along with promoter constructs, were transfected into T4-2 cells that overexpressed HOXD10. Transfection was performed with 1 µl XFect transfection reagent (Clontech, cat# 631318), 1.5 µg of promoter DNA and 200 nM of decoy oligonucleotides according to the manufacturer's protocol. Cells were harvested at 48 hr post transfection. The luciferase activity was analyzed using the Bright-Glo Luciferase assay system (E2610, Promega) according to the manufacturer's protocol, and the activity was normalized using protein concentration.

## Analysis of transcription factor (TF) binding sites

TF binding sites within the promoter regions were predicted by AlGGEN PROMO software (http://alggen.lsi.upc.es/cgi-bin/promo_v3/promo/promoinit.cgi?dirDB=TF_8.3) (*Farré et al., 2003*). The feasibility of these predicted sites was indicated as the 'Dissimilarity' to the canonical sequence (0% as the best match). The significance of the predicted site was indicated as the 'Frequency' in the genomic background ('Random Expectancy' (RE) value x $10^{-3}$) (*Farré et al., 2003*).

## Chromatin Immunoprecipitation (ChIP)

ChIP assays were performed as described by *Saccani et al. (2001)* with a minor modification. Cells were plated at $2 \times 10^6$/100-mm plate and maintained overnight. Then, cells were maintained in the fresh medium containing 5% Matrigel for 24 hr (for T4-2 Rev, 350 nM AG1478 was added). Cells placed in fresh medium with 1% formaldehyde for 10 min, scraped off from the dish with PBS and processed for nuclear extraction. Chromatins were sonicated to ~500 bp fragments and

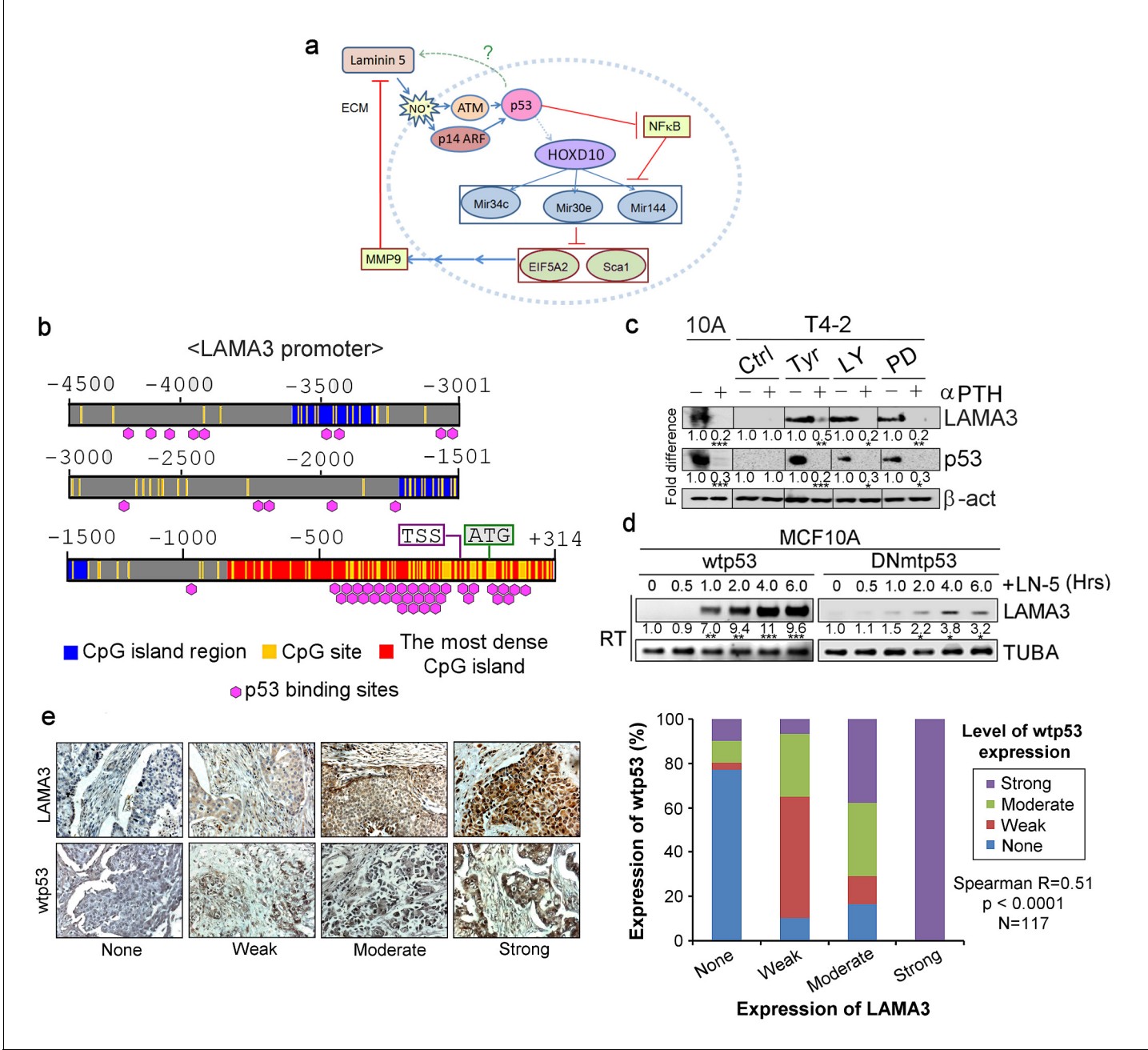

**Figure 9.** Activated p53 in turn upregulates expression of the endogenous alpha chain of LN5. (**a**) A schematic of the morphogenetic loop dissected to date with a predicted feedback loop between p53 and LN5. (**b**) The promoter region of the *LAMA3* gene contained the CpG island that harbors numerous p53-binding sites around the transcription start site (TSS). (**c**) Representative result of western analysis (n = 3) for LAMA3 expression in T4-2 cells treated with a reverting agent (AG1478, LY294002 or PD98059) in the absence or presence of a p53 inhibitor, α-PFT. β-actin serves as a loading control. Fold difference was determined with respect to the Ctrl T4-2. *p<0.05; **p<0.01 and ***p<0.001. For each analysis, replicate experiments (n = 3) were performed, and representative data are shown. (**d**) Representative result of RT-PCR analysis (n = 3) showing that exogenous LN5 (1 μg/ml) upregulated *LAMA3* transcription in MCF10A cells expressing wild-type p53, but not in cells expressing DNp53. α-tubulin (TUBA) was used as a control. Fold difference was determined with respect to time 0. *p<0.05; **p<0.01 and ***p<0.001. (**e**) (Left) Representative images of the IHC staining of breast cancer tissues (n = 117) for LAMA3 and wild-type p53. (Right) Correlation analysis between LAMA3 expression and wild-type p53 expression in breast tumors.

DOI: https://doi.org/10.7554/eLife.26148.032

The following figure supplement is available for figure 9:

**Figure supplement 1.** LAMA3 expression depends on p53 activation.
DOI: https://doi.org/10.7554/eLife.26148.033

**Table 8.** LAMA3 promoter regions harboring binding sites of p53.

| Promoter | TXN bound | Start nt from TSS | End nt from TSS | String | Dissimilarity (%) | Frequency (random expectancy x $10^{-3}$) | |
|---|---|---|---|---|---|---|---|
| | | | | | | Equally | Query |
| LAMA3 | p53 | −4245 | −4239 | TGAGCCC | 8.8 | $2 \times 10^{-3}$ | $2 \times 10^{-3}$ |
| | | −4143 | −4137 | GGGCAGA | 1.7 | $9 \times 10^{-4}$ | $8 \times 10^{-4}$ |
| | | −4063 | −4057 | TCTGCCC | 1.7 | $9 \times 10^{-4}$ | $8 \times 10^{-4}$ |
| | | −3597 | −3591 | GGTGCCC | 4 | $1 \times 10^{-3}$ | $1 \times 10^{-3}$ |
| | | −3585 | −3579 | CACGCCC | 3.3 | $1 \times 10^{-3}$ | $1 \times 10^{-3}$ |
| | | −3451 | −3445 | GGCGCCC | 7.4 | $1 \times 10^{-3}$ | $1 \times 10^{-3}$ |
| | | 813 | 819 | ACTGCCC | 3.5 | $1 \times 10^{-3}$ | $1 \times 10^{-3}$ |
| | | −4547 | −4541 | CTTGCCC | 0.2 | $9 \times 10^{-4}$ | $7 \times 10^{-4}$ |
| | | −3094 | −3088 | TGAGCC | 6.7 | $2 \times 10^{-3}$ | $2 \times 10^{-3}$ |
| | | −2804 | −2798 | CACGCCC | 3.3 | $1 \times 10^{-3}$ | $1 \times 10^{-3}$ |
| | | −2718 | −2712 | TCTGCCC | 1.7 | $9 \times 10^{-4}$ | $8 \times 10^{-4}$ |
| | | −2347 | −2341 | CCAGCCC | 3.7 | $1 \times 10^{-3}$ | $1 \times 10^{-3}$ |
| | | −2328 | −2322 | GGGCTCT | 8.5 | $3 \times 10^{-4}$ | $2 \times 10^{-4}$ |
| | | −1914 | −1908 | GTCGCCC | 6.4 | $1 \times 10^{-3}$ | $1 \times 10^{-3}$ |
| | | −1798 | −1792 | ACCGCCC | 6.8 | $2 \times 10^{-3}$ | $2 \times 10^{-3}$ |
| | | −948 | −942 | GTCGCCC | 6.4 | $1 \times 10^{-3}$ | $9 \times 10^{-4}$ |
| | | −437 | −431 | TCTGCCC | 1.7 | $9 \times 10^{-4}$ | $8 \times 10^{-4}$ |
| | | −420 | −414 | GGGCGGC | 6.1 | $1 \times 10^{-3}$ | $1 \times 10^{-3}$ |
| | | −374 | −368 | GGGCGGC | 3.5 | $1 \times 10^{-3}$ | $1 \times 10^{-3}$ |
| | | −355 | −349 | GGGCGCG | 4.6 | $6 \times 10^{-4}$ | $4 \times 10^{-4}$ |
| | | −342 | −336 | CTGGCCC | 4.3 | $6 \times 10^{-4}$ | $4 \times 10^{-4}$ |
| | | −325 | −319 | GGGCCGC | 6.9 | $2 \times 10^{-3}$ | $2 \times 10^{-3}$ |
| | | −314 | −308 | GGGCGGG | 3.3 | $1 \times 10^{-3}$ | $1 \times 10^{-3}$ |
| | | −310 | −304 | GGGCAGG | 0 | $9 \times 10^{-4}$ | $7 \times 10^{-4}$ |
| | | −296 | −290 | GGGCACA | 3 | $1 \times 10^{-3}$ | $1 \times 10^{-3}$ |
| | | −256 | −250 | GCAGCCC | 6.5 | $1 \times 10^{-3}$ | $9 \times 10^{-4}$ |
| | | −236 | −230 | TCAGCCC | 5.5 | $1 \times 10^{-3}$ | $1 \times 10^{-3}$ |
| | | −223 | −217 | TCTGCCC | 1.7 | $9 \times 10^{-4}$ | $8 \times 10^{-4}$ |
| | | −183 | −177 | TCAGCCC | 5.5 | $1 \times 10^{-3}$ | $1 \times 10^{-3}$ |
| | | −1127 | −1121 | GGGCGCC | 7.4 | $1 \times 10^{-3}$ | $1 \times 10^{-3}$ |
| | | −126 | −120 | GGCGCCC | 7.4 | $1 \times 10^{-3}$ | $1 \times 10^{-3}$ |
| | | −96 | −90 | GGGCCAA | 6 | $1 \times 10^{-3}$ | $1 \times 10^{-3}$ |
| | | −85 | −79 | GGGCGGG | 3.3 | $1 \times 10^{-3}$ | $1 \times 10^{-3}$ |
| | | −75 | −69 | GGGCGGG | 3.3 | $1 \times 10^{-3}$ | $1 \times 10^{-3}$ |
| | | −70 | −64 | GGGCGGG | 3.3 | $1 \times 10^{-3}$ | $1 \times 10^{-3}$ |
| | | −65 | −59 | GGGCGCA | 6.4 | $1 \times 10^{-3}$ | $1 \times 10^{-3}$ |
| | | −40 | −34 | GGGCGGC | 6.1 | $1 \times 10^{-3}$ | $1 \times 10^{-3}$ |
| | | −24 | −18 | GGGCGGC | 6.1 | $1 \times 10^{-3}$ | $1 \times 10^{-3}$ |
| | | 1 | 7 | GGGCCAG | 4.3 | $6 \times 10^{-4}$ | $4 \times 10^{-4}$ |
| | | 7 | 13 | GGGCAGC | 2.8 | $1 \times 10^{-3}$ | $1 \times 10^{-3}$ |
| | | 45 | 51 | GGGCGCG | 4.6 | $6 \times 10^{-4}$ | $4 \times 10^{-4}$ |
| | | 101 | 107 | GGGCGTG | 3.3 | $1 \times 10^{-3}$ | $1 \times 10^{-3}$ |
| | | 155 | 161 | TGAGCCC | 6.7 | $2 \times 10^{-3}$ | $2 \times 10^{-3}$ |

*Table 8 continued on next page*

*Table 8 continued*

| Promoter | TXN bound | Start nt from TSS | End nt from TSS | String | Dissimilarity (%) | Frequency (random expectancy x $10^{-3}$) | |
|---|---|---|---|---|---|---|---|
| | | | | | | Equally | Query |
| | | 160 | 166 | CCGGCCC | 4.1 | $1 \times 10^{-3}$ | $1 \times 10^{-3}$ |
| | | 216 | 222 | GGGCGGG | 3.3 | $1 \times 10^{-3}$ | $1 \times 10^{-3}$ |
| | | 200 | 206 | GGGCGGG | 3.3 | $1 \times 10^{-3}$ | $1 \times 10^{-3}$ |
| | | 206 | 212 | GGGCGGC | 6.1 | $1 \times 10^{-3}$ | $1 \times 10^{-3}$ |
| | | 220 | 226 | AAAGCCC | 7.2 | $1 \times 10^{-3}$ | $1 \times 10^{-3}$ |
| | | 236 | 242 | GGGCTGC | 6.5 | $1 \times 10^{-3}$ | $1 \times 10^{-3}$ |
| | | 251 | 257 | GGGCGCG | 4.6 | $6 \times 10^{-4}$ | $4 \times 10^{-4}$ |

DOI: https://doi.org/10.7554/eLife.26148.034

immunoprecipitated with control rabbit IgG, HOXD10 and p65 antibodies at 4°C overnight. Chromatin-antibody complexes were washed with buffer 1 [0.1% SDS, 0.5% Triton X-100, 2 mM EDTA, 20 mM Tris-HCl (pH 8.0), 150 mM NaCl], buffer 2 [0.1% SDS, 2 mM EDTA, 20 mM Tris-HCl (pH 8.0), 500 mM NaCl] then TE buffer [10 mMTris-HCl (pH 8.0) 1 mM EDTA]. After reversal of cross-linking by heating at 65°C overnight, immunoprecipitated chromatin was subjected to PCR reaction for ~300 bp fragments around HOXD10/NFκB binding sites in miRNA promoters (miR-34c: −1 ~ 0 kb, miR-30e: −3~−2 kb, miR-144: −3~−2 kb) with the appropriate primers (*Table 9*).

## Protein array

The relative abundance of the secreted laminin chains was determined with ImmunoCruz Cell Adhesion-2 MicroArray (sc-200006, Santa Cruz Biotechnologies, Santa Cruz, CA, USA) according to the manufacturer's protocol. Briefly, cells were plated at $2 \times 10^{6}$/100-mm plate and maintained overnight. Cells were maintained in the fresh medium containing 5% Matrigel for 24 hr. The CM was harvested and spun to remove the Matrigel drip. The medium was concentrated to 1 ml using Amicon Ultra-15 centrifugal filter units (3 kDa cut off, Millipore). The protein concentration was determined with DC Protein Assay reagent (Bio-Rad) and normalized to 1 mg/ml. 250 μg protein was labeled with Cy3 dye (Cy3 Mono-Reactive Dye Pack, GE Healthcare, Milwaukee, WI, USA). The labeled protein was dissolved in 1.5 ml desalting buffer, and unbound dye was removed by using Amicon Ultra-15 centrifugal filter units that concentrated the protein to 500 μl. The labeled protein was hybridized with array slides, and slides were scanned and analyzed by the CruzScan Scanning service (sc-200215, Santa Cruz).

## Immunohistochemistry

Breast cancer tissue arrays containing 150 paraffin-embedded sections of normal and malignant tissues with pathological information (stages I through III) were obtained from US Biomax, Inc (BR1503b). Slides were deparaffinized, hydrated, and treated with antigen unmasking solutions (Vector Laboratories, Inc.). After being blocked with 0.3% $H_2O_2$ and nonimmune goat serum, sections were incubated at room temperature with an antibody against S-nitrosocysteine (Abcam, Cambridge, MA, USA, clone HY8E12), human LAMA3 (R&D Systems, Minneapolis, MN, USA, , clone 546215) or wild-type human p53 (EMD Millipore, clone pAb1620) and link antibodies, followed by peroxidase-conjugated streptavidin complex and diaminobenzidine tetrahydroxy chloride solution as the peroxidase substrate (Vector Laboratories, Burlingame, CA, USA). The sections were counter-stained with hematoxylin. Photomicrographs were taken with the Zeiss Axioskop Imaging platform and Axion Vision software (Version 4.7).

## MMP-9 measurement

MMP-9 secreted into CM was measured using the MMP-9 ELISA Kit (ThermoFisher Scientific) according to the manufacturer's protocol. Assay samples were prepared in the dark. Briefly, cells were plated at $1 \times 10^{6}$/60-mm plate and maintained overnight. Cells were maintained in 2 ml of the fresh medium containing 5% Matrigel for 24 hr. The CM was harvested and spun to remove the

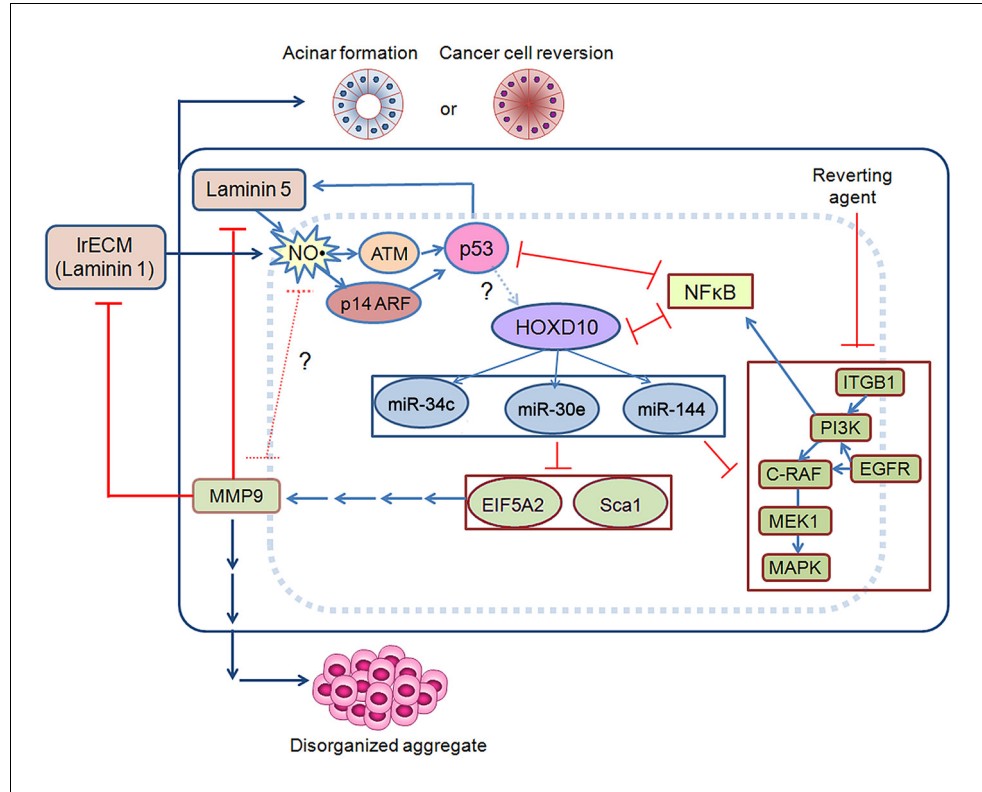

**Figure 10.** Schematic for acinar morphogenesis and phenotypic reversion of tumor cells in response to LN1 or LN5.

DOI: https://doi.org/10.7554/eLife.26148.035

Matrigel drip. The cleared CM was diluted 100-fold and analyzed for MMP-9 concentration using MMP-9 standards based on the optical density values at 450 nm.

## Nitrite/nitrate measurement

To quantify the cumulative level of NO produced, the more stable oxidation product nitrite/nitrate was measured using the Measure-IT High-Sensitivity Nitrite Assay Kit (ThermoFisher Scientific) according to the manufacturer's protocol. Assay samples were prepared in the dark. Briefly, cells were plated at $1 \times 10^6$/60-mm plate and maintained overnight. Cells were maintained in 2 ml of the fresh medium containing 5% Matrigel for the designated time periods. The CM was harvested and spun to remove the Matrigel drip. 10 µl of the cleared CM was analyzed for nitrite concentration using nitrite standards at the excitation/emission maxima of 340/410 nm.

## Detection of NO production in live cells

To capture a snap shot of NO level in live cells after laminin addition, a dye DAF-FM DA (4-amino-5-methylamino-2',7'-difluorofluorescein diacetate, ThermoFisher Scientific) was used according to the manufacturer's protocol. The signal intensity/area/cell was measured with ImageJ.

## Breast tissues and *ex vivo* 3D organoid cultures

Breast tissues from reduction mammoplasties were obtained from the Cooperative Human Tissue Network (CHTN), a program funded by the National Cancer Institute. All specimens were collected with patient consent and were reported negative for proliferative breast disease by board-certified pathologists. Use of these anonymous samples was granted exemption status by the University of California at Berkeley Institutional Review Board according to the Code of Federal Regulations 45 CFR 46.101. Upon receipt, the tissues were rinsed with PBS, minced and incubated overnight with 0.1% collagenase as previously described (with gentle agitation) (*Hines et al., 2015*). The resulting

**Table 9.** List of oligonucleotide sequences of molecules listed in the manuscript.

**Northern probes**

| | |
|---|---|
| miR-450b-5p | 5′-TAT TCA GGA ACA TAT TGC AAA A-3′ |
| miR-495 | 5′-AAG AAG TGC ACC ATG TTT GTT T-3′ |
| miR-30e | 5′-CTT CCA GTC AAG GAT GTT TAC A-3′ |
| miR-330–3 p | 5′-TCT CTG CAG GCC GTG TGC TTT GC-3′ |
| miR-382 | 5′-CGA ATC CAC CAC GAA CAA CTT C-3′ |
| miR-423–3 p | 5′-ACT GAG GGG CCT CAG ACC GAG CT-3′ |
| miR-135a | 5′-TCA CAT AGG AAT AAA AAG CCA TA-3′ |
| miR-144 | 5′-AGT ACA TCA TCT ATA CTG TA-3′ |
| miR-301b | 5′-GCT TTG ACA ATA TCA TTG CAC TG-3′ |
| miR-590–3 p | 5′-ACT AGC TTA TAC ATA AAA TTA-3′ |
| miR-301a | 5′-GCT TTG ACA ATA CTA TTG CAC TG-3′ |
| miR-34c-5p | 5′-GCA ATC AGC TAA CTA CAC TGC CT-3′ |

**RT-PCR**

| | |
|---|---|
| EIF5A2 | |
| FW | 5′-ATG GCA GAC GAA ATT GAT TTC ACT A-3′ |
| RV | 5′-CTC ATT GCA CAC ATG ACA GAC-3′ |
| SCA1 | |
| FW | 5′-ACG GTC ATT CAG ACC ACA CA-3′ |
| RV | 5′-CAG GGT TGA AGT TCT CGC TC-3′ |
| ITGB1 | |
| FW | 5′-CGC CGC GCG GAA AAG ATG AAT-3′ |
| RV | 5′-TGG GCT GGT GCA GTT CTG TTC A-3′ |
| c-RAF | |
| FW | 5′-CGA CCC ACA GTG GAC GAT CCA G-3′ |
| RV | 5′-AGA TAA TGC TGG CCG ACT GGC CT-3′ |
| MEK1 | |
| FW | 5′-AAG GGA ATC CCG GGC TGC CGA A-3′ |
| RV | 5′-GCC ATC GCT GTA GAA CGC ACC A-3′ |
| MAPK | |
| FW | 5′-GCA CCG TGA CCT CAA GCC TTC-3′ |
| RV | 5′-CAC CGA TGT CTG AGC ACG TCC AG-3′ |
| LAMA3 | |
| FW 5′-GAT GGC TCA GGC ATA TGT GTT-3′ | |
| RV 5′-CTG GCC ATT GCT GTT ACA ACT-3′ | |
| TUBA | |
| FW | 5′-TGA CCT GAC AGA ATT CCA GAC CA-3′ |
| RV | 5′-GCA TTG ACA TCT TTG GGA CC AC-3′ |

**shRNA** (target sequence underlined; *Bam*H1/*Eco*R1 cohesive ends italicized)

| | |
|---|---|
| EIF5A2sh | |
| Sense | 5′-*GAT* CCG <u>CTG CCA GAA GGT GAA CTA G</u>CT TCC TGT CAG ATA TAT CTC TCC TTC CAC ACT TTT T*G*-3′ |
| Antisense | 5′-*AAT TC*A AAA A<u>CT GCC AGA AGG TGA ACT AG</u>T CTG ACA GGA AGT ATA TCT CTC CTT CCA CAC *G*-3′ |
| Sca1sh | |

*Table 9 continued on next page*

*Table 9 continued*

### Northern probes

| | |
|---|---|
| Sense | 5′-*GAT CCG* AAC CTG AAG AAC GGC TCT CTT<br>CCT GTC AGA AGA GCC GTT CTT CAG GTT CTT TTT *G*-3′ |
| Antisense | 5′-*AAT TCA* AAA AGA ACC TGA AGA ACG GCT CTT CTG<br> ACA GGA AGA GAG CCG TTC TTC AGG TTC *G*-3′ |

### p65sh

| | |
|---|---|
| Sense | 5′-*GAT CCG* GAC ATA TGA GAC CTT CAA CTT<br>CCT GTC AGA TTG AAG GTC TCA TAT GTC CTT TTT *G*-3′ |
| Antisense | 5′-*AAT TCA* AAA AGG ACA TAT GAG ACC TTC AAT<br>CTG ACA GGA AGT TGA AGG TCT CAT ATG TCC *G*-3′ |

### p50/p100sh

| | |
|---|---|
| Sense | 5′-*GAT CCG* AGC TAA TCC GCC AAG CAG CTT<br>CCT GTC AGA CTG CTT GGC GGA TTA GCT CTT TTT *G*-3′ |
| Antisense | 5′-*AAT TCA* AAA AGA GCT AAT CCG CCA AGC AGT CTG ACA G<br>GA AGC TGC TTG GCG GAT TAG CTC *G*-3′ |

### LAMA3sh

| | |
|---|---|
| Sense | 5′-*GAT CCG* GAG TCC TTC TGG ATT ACC CTT CCT<br>GTC AGA GGT AAT CCA GAA GGA CTC CTT TTT *G*-3′ |
| Antisense | 5′-*AAT TCA* AAA AGG AGT CCT TCT GGA TTA CCT CTG<br>ACA GGA AGG GTA ATC CAG AAG GAC TCC *G*-3′ |

### Overexpressing constructs

#### HOXD10

| | |
|---|---|
| FW | 5′-CGG CAG *GCG CGC* CGC CAC CAT GTC CTT TCC CAA CAG CTC TCC T-3′ (*Asc*I site italicized) |
| RV | 5′-CCG GCC *GAA TTC* CTA AGA AAA CGT GAG GTT GGC GGT CAG-3′ (*EcoR*1 site italicized) |

#### p53

| | |
|---|---|
| FW | 5′-GAT CTC *GGA TCC* GCC ACC ATG GAG GAG CCG CAG TCA GAT CCT AGC-3′ (*BamH*1 site italicized) |
| RV | 5′-TAC AG*G AAT TC*T CAG TCT GAG TCA GGC CCT TCT GTC TTG AAC ATG-3′ (*EcoR*1 site italicized) |

#### ATXN1 and EIF5A2

| | |
|---|---|
| FW | 5′-TCT AGA GCC CGG GCG CGC CGG CCG CCG CGA TCG CCA TG-3′ |
| RV | 5-″GCA GAT CCT TCG CGG CCG CGT TAA CCT TA<br> TCG TCG TCA TCC TTG TAA TCC AGG ATA TCA TTT GC-3′ |

### miRNA reporter constructs (*Mlu*1/*Xho*1 sites italicized)

#### miR-34c

##### 3–0 kb

| | |
|---|---|
| FW | 5′-GAC T*AC GCG T*AC CGC TGG CAG TTC ATT TTA GCT C-3′ (*Mlu*1 site italicized) |
| RV | 5′-GAC T*CT CGA G*CT AGA AGA TGG AGG CCC AGA TTC TTG AGA C-3′ (*Xho*1 site italicized) |

##### 2–0 kb

| | |
|---|---|
| FW | 5′-GAC T*AC GCG T*CT TGG CTT CCT CCT AGT CAT CAA CCT-3′ (*Mlu*1 site italicized) |
| RV | 5′-GAC T*CT CGA G*TC TGA TCT AGC AGG AGG GAC AAA GAG-3′ (*Xho*1 site italicized) |

##### 1–0 kb

| | |
|---|---|
| FW | 5′-GAC T*AC GCG T*TC CCT TCA CTA TGG GGT GTA CAG AAC-3′ (*Mlu*1 site italicized) |
| RV | 5′-GAC T*CT CGA G*CT AGA AGA TGG AGG CCC AGA TTC TTG AGA C-3′ (*Xho*1 site italicized) |

##### 3–2 kb

| | |
|---|---|
| FW | 5′-GAC T*AC GCG T*TT ATA AAA ACC GCT GGC AGT TCA TTT TAG C-3′ (*Mlu*1 site italicized) |
| RV | 5′-GAC T*CT CGA G*AG GAG GAA GCC AAG AAG AGT GTA GAA AAC A-3′ (*Xho*1 site italicized) |

##### 2–1 kb

| | |
|---|---|
| FW | 5′-GAC T*AC GCG T*CT ATT CTC CCA CCT CAG CC TCC AAG TAG-3′ (*Mlu*1 site italicized) |
| RV | 5′-GAC T*CT CGA G*CT GTA CAC CCC ATA GTG AAG GGA AAG AAA C-3′ (*Xho*1 site italicized) |

*Table 9 continued*

**Northern probes**

| | |
|---|---|
| miR-30e | |
| 3–0 kb | |
| FW | 5′-GAC T*AC GCG T*GC CAC CAT GCC CGG CTA A-3′ (*Mlu*1 site italicized) |
| RV | 5′-GAC T*CT CGA* GGG GAG CTC GAG ATC TGA GTT TTG ACC-3′ (*Xho*1 site italicized) |
| 2–0 kb | |
| FW | 5′-GAC T*AC GCG T*CT GGT CTT GAA CTC CTG ACC TCG TCA T-3′ (*Mlu*1 site italicized) |
| RV | 5′-GAC T*CT CGA* GTT CGG GAG CTC GAG ATC TGA GTT TTG-3′ (*Xho*1 site italicized) |
| 1–0 kb | |
| FW | 5′-GAC T*AC GCG T*TT AGA TCT GGG TAC AGA TGA AGG AAT TGA GAC TCC-3′ (*Mlu*1 site italicized) |
| RV | 5′-GAC T*CT CGA* GTT CGG GAG CTC GAG ATC TGA TGG TTG-3′ (*Xho*1 site italicized) |
| 3–2 kb | |
| FW | 5′-GAC T*AC GCG T*CT TTT TGA ACT CCA GCA GCA CAT GAA CTA T-3′ (*Mlu*1 site italicized) |
| RV | 5′-GAC T*CT CGA* GGG CCT TGT TTT GAC CAA TGA AAT ATG AGT A-3′ (*Xho*1 site italicized) |
| 2–1 kb | |
| FW | 5′-GAC T*AC GCG T*CT GGT CTT GAA CTC CTG ACC TCG TCA T-3′ (*Mlu*1 site italicized) |
| RV | 5′-GAC T*CT CGA* GAC ACT TGA CTT CAG GGA GTC TCA ATT CCT T-3′ (*Xho*1 site italicized) |
| miR-144 | |
| 3–0 kb | |
| FW | 5′-GAC T*AC GCG T*CT CAC TAT AAG ACT CGG GCC AAG CAC TTC-3′ (*Mlu*1 site italicized) |
| RV | 5′-GAC T*CT CGA* GGC CAG TTG TGG TGG CAT GTG-3′ (*Xho*1 site italicized) |
| 2–0 kb | |
| FW | 5′-GAC T*AC GCG T*GT TGC CCA GGC TGG AGT ACA ATA GGA T-3′ (*Mlu*1 site italicized) |
| RV | 5′-GAC T*CT CGA* GAA TTA GCC AGT TGT GGT GGC ATG TG-3′ (*Xho*1 site italicized) |
| 1–0 kb | |
| FW | 5′-GAC T*AC GCG T*GT ACT GGG GAG GCA GAG GAA TGG AAG-3′ (*Mlu*1 site italicized) |
| RV | 5′-GAC T*CT CGA* GAA TTA GCC AGT TGT GGT GGC ATG TG-3′ (*Xho*1 site italicized) |
| 3–2 kb | |
| FW | 5′-GAC T*AC GCG T*CC TAT TCC TAG CGG GTT TGT GCA TAG AG-3′ (*Mlu*1 site italicized) |
| RV | 5′-GAC T*AG ATC T*CT GGG CAA CAA GAG CAA AAC TGG ATC-3′ (*Bgl*1I site italicized) |
| 2–1 kb | |
| FW | 5′-GAC T*AC GCG T*CC CAG GCT GGA GTA CAA TAG GAT GAT CT-3′ (*Mlu*1 site italicized) |
| RV | 5′-GAC T*CT CGA* GGC CCA GGG CTG TTT CCT GGA TA TT-3′ (*Xho*1 site italicized) |
| ChIP | |
| miR-34c (−1~0 kb) | |
| FW | 5′-GTG TCA GCA ATG GGT GCT CTA-3′ |
| RV | 5′-CCA GAG GAG GTG AGA CTT GAG-3′ |
| miR-30e (−3~−2 kb) | |
| FW | 5′-GAG GCA GTC TGA GAT ATT CCC-3′ |
| RV | 5′-CTG CAG CAT AAC ATG CTA GCT-3′ |
| miR-144 (−3~−2 kb) | |
| FW | 5′-CTG TGA TGA GGA CAA CAG TAA-3′ |
| RV | 5′-ATC CCC CTA CCT CAG CCT CTC-3′ |

DOI: https://doi.org/10.7554/eLife.26148.036

divested tissue fragments (organoids) were rinsed with PBS and collected by centrifugation (100 g × 2 min). Lactiferous ducts and terminal ductal lobular units (TDLU) were individually isolated using a micromanipulator and drawn glass needles using a screw-actuated micrometer driven hamilton syringe for suction/injection pressure. Single organoids were subsequently embedded in 50% growth factor reduced Matrigel (BD Biosciences) and overlayed with M87 growth medium. At 2 hr post seeding, medium was refreshed with L-NAME (NO inhibitor) containing medium at 5 mM. Cells were incubated at 37°C/5% $CO_2$. Medium was refreshed every other day for the length of the experiment (14 d).

## Live cell imaging and cell tracking

Three-dimensional live cell imaging was performed using a Zeiss LSM 710 Meta confocal microscope and Zen Version 8.1 software. Cells were mixed with lrECM, seeded and covered along with complete growth media in a Lab-Tek 4-well chambered coverglass 2 hr prior to image capturing. Samples were placed in a 37 °C humidified microscope stage incubator with 5% $CO_2$. Images of 512 × 512 pixels in XY coordinates with a maximum Z-axis displacement of 75 μm were acquired using a 0.8 NA 20 × air objective at one frame/second. Images were captured successively at 20 min intervals for 48 hr. Samples were simultaneous excited by the 488 nm light (argon ion laser) at a power of <3% maximum and 546 nm light (a solid-state laser) at a power of <10% maximum. A secondary dichroic mirror was used in the emission pathway to separate the red (band-pass filters 560–575 nm) and green (band-pass filters 505–525 nm) channels. Gain was set between 100 and 180. Processed data were imported into Imaris (Bitplane, South Windsor, CT, USA), and nuclei were modeled (detection diameter: 5,800~6,500 nm). The nuclei were tracked over time using the tracking function of Imaris with the maximum distance of 2,500–20,000 nm and the maximum gap size of 1.

## Statistics

Unless otherwise indicated, statistical analyses were performed using Graph Pad Prism Version 5 software and an unpaired two-tailed Student's t-test for parametric tests and Spearman correlation analysis for non-parametric tests. P-values of 0.05 or less were considered significant. Average results of multiple experiments (n > 3) are presented as the arithmetic mean ± SEM.

## Acknowledgements

We thank all members of the Bissell Laboratory for constructive comments. In particular, we would like to thank Dr Curt Hines, Kremena Karagyozova and Kate Thi for providing normal human breast tissues and preparing ex vivo organotypic cultures. We also would like to thank Drs. Eva Lee and Alvin Lo in the Bissell laboratory and Samantha Metzger and Matthew Bommarito in the Furuta laboratory for their excellent technical support. The authors thank Dr Sun-Young Lee for helping with accuracy of the galley and additional suggestions for improvement. The authors thank Sarah J Lee for her excellent administrative assistance with the many changes required during the final stages of manuscript preparation and galley proofing. This work was supported by grants from the US Department of Defense Innovator Expansion Award (W81XWH0810736); Breast Cancer Research Foundation (BCRF); multiple grants from the NCI (R01CA064786; U01CA143233, U54CA112970 and U54CA143836 – Bay Area Physical Sciences Oncology Center) to MJB; and in part by the University of Toledo Department of Cancer Biology Startup Fund and an Ohio Cancer Research Grant (Project #: 5017) to SF.

## Additional information

### Funding

| Funder | Grant reference number | Author |
| --- | --- | --- |
| Ohio Cancer Research | 5017 | Saori Furuta |
| National Cancer Institute | R01CA064786 | Mina J Bissell |
| U.S. Department of Defense | W81XWH0810736 | Mina J Bissell |

| Breast Cancer Research Foundation | | Mina J Bissell |
|---|---|---|
| National Cancer Institute | U01CA143233 | Mina J Bissell |
| National Cancer Institute | U54CA112970 | Mina J Bissell |
| National Cancer Institute | U54CA143836 | Mina J Bissell |

The funders had no role in study design, data collection and interpretation, or the decision to submit the work for publication.

### Author contributions
Saori Furuta, Conceptualization, Data curation, Formal analysis, Funding acquisition, Investigation, Visualization, Methodology, Writing—original draft, Project administration, Writing—review and editing; Gang Ren, Data curation, Performed the experiments for revision, Acquired and analysed data; Jian-Hua Mao, Data curation, Formal analysis, Writing—review and editing, Performed statistical analyses and provided primary tissues; Mina J Bissell, Conceptualization, Resources, Supervision, Funding acquisition, Writing—original draft, Project administration, Writing—review and editing

### Author ORCIDs
Saori Furuta (iD) https://orcid.org/0000-0003-1121-0487
Mina J Bissell (iD) http://orcid.org/0000-0001-5841-4423

### Decision letter and Author response
Decision letter https://doi.org/10.7554/eLife.26148.043
Author response https://doi.org/10.7554/eLife.26148.044

# Additional files

### Supplementary files
• Transparent reporting form
DOI: https://doi.org/10.7554/eLife.26148.037

### Major datasets
The following previously published dataset was used:

| Author(s) | Year | Dataset title | Dataset URL | Database, license, and accessibility information |
|---|---|---|---|---|
| Becker-Weimann S, Bissell MJ, Onodera Y, Rizki A | 2013 | Gene expression in organized and disorganized human breast epithelial cells | https://www.ncbi.nlm.nih.gov/geo/query/acc.cgi?acc=GSE50444 | Publicly available at the NCBI Gene Expression Omnibus (accession no: GSE50444) |

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
