## [Decision Letter]

Thank you for submitting your article "Laminins signal to initiate the reciprocal loop that informs breast morphogenesis by activating NO, p53 and microRNAs" for consideration by *eLife*. Your article has been reviewed by two peer reviewers, and the evaluation has been overseen by a Reviewing Editor and Fiona Watt as the Senior Editor. The following individuals involved in review of your submission have agreed to reveal their identity: David Bryant (Reviewer #3).

The reviewers have discussed the reviews with one another and the Reviewing Editor has drafted this decision to help you prepare a revised submission.

Summary:

This manuscript addresses the activation of p53 during mammary acinar formation upon laminin induced NO production. They also study the feedback regulation of laminin expression either directly via p53 driven transcription, or indirectly via protein stability modulated by p53-Hoxd10/Nfkb-miR34c/30e/144-EIF5A2/SCA1-MMP9. The overall findings are corroborated by experimental support, and will be of interest to the broader readership of e*Life*. A few concerns below however need to be addressed prior to publication.

Essential revisions:

1) Can the authors analyze gene expression changes shortly after overexpression of the miRs and then look for immediate changes in mRNAs harboring miR binding sites to identify putative direct targets? Alternatively, can EIF5A2/SCA1 re-expression without predicted miR-binding-sites suppress the reversion conferred by miRs overexpression? This at least would suggest that these are the major relevant targets in this setting.

2) The regulation of miR promoter activity by HOXD10/NFKB lacks a critical control. The authors need to mutate the putative binding sites and then test for reporter activity. A related critical issue is whether these observed TF motifs are significantly enriched in putative target promoter regions compared to genomic background where these sequences could be frequently found. Similarly in the P53 regulation of laminin transcription experiment: is the occurrence of P53 motifs enriched over genomic background?

3) The Introduction extensively describes the results. This is somewhat repetitive with the Results section. Revision of the Introduction would focus readers to the important concepts.

---

## [Author Response]

Essential revisions:1) Can the authors analyze gene expression changes shortly after overexpression of the miRs and then look for immediate changes in mRNAs harboring miR binding sites to identify putative direct targets? Alternatively, can EIF5A2/SCA1 re-expression without predicted miR-binding-sites suppress the reversion conferred by miRs overexpression? This at least would suggest that these are the major relevant targets in this setting.

We agree with both suggestions by the reviewer. Between the two experiments, we chose the second experiment because of the technical difficulty to perform the first experiment. Using T4-2 breast cancer cells, which are hard to transfect, it is difficult to analyze gene expression changes shortly after overexpression of the miRNAs. In fact, to overexpress miRNAs in T4-2 cells, we had to employ lentiviral system and select positive clones, which usually took several weeks.

Thus, we performed the second experiment that validates that EIF5A2 and SCA1 are the major relevant target of the miRNAs by restoring their expression in miRNA-expressing cells. The 3 miRNAs bind EIF5A2 and SCA1 transcripts at the 3’UTR (Table 5). Using lentivirus, we overexpressed cDNAs of EIF5A2 and SCA1 in T4-2 cells expressing the miRNAs. These cDNAs do not contain the 3’UTR, and thus are unbound by the miRNAs (Author response image 1). Expression of EIF5A2 and SCA1 cDNA’s were confirmed by western analysis (Figure 3—figure supplement 1). Restoration of EIF5A2 and SCA1 in miRNA-expressing T4-2 cells severely impaired tumor reversion (Figure 3—figure supplement 1). This result confirms that downmodulation of EIF5A2 and SCAI plays major roles in tumor reversion mediated by the 3 miRNAs.

**Author response image 1. respfig1:** Scheme of overexpression of miRNA target gene (SCAI/EIF5A2) without miRNA binding sites in the 3’UTR. (A, B) T4-2 cells overexpressing each of the three miRNAs (Mir-34c, Mir-30e, and Mir-144) are generated. (C) These miRNA-overexpressing cells will be transfected with SCAI/EIF5A2 cDNA (no 3’UTR, no miRNA binding sites).

2) The regulation of miR promoter activity by HOXD10/NFKB lacks a critical control. The authors need to mutate the putative binding sites and then test for reporter activity.

We appreciate the reviewer’s comment and have performed the suggested experiment. We utilized the decoy technology to validate HOXD10 and NF-κB binding at the predicted sites within miRNA promoters (Author response image 2)(4). Here, the wild-type decoy, a double strand of a putative binding sequence of HOXD10 or NF-κB in each of the miRNA promoters (miR34c, miR30e or miR144), was overexpressed to sequester the respective transcription factor. In comparison, the mutant decoy carrying point mutations at the core binding sequence was expressed in the same cells. A pool of NFkB decoys (SCR, WT or MT) for each miRNA promoter was expressed in Ctrl T4-2 cells (high NFkB). For T4-2 cells overexpressing HOXD10 (high HOXD10), a pool of HOXD10 decoys (SCR, WT or MT) for each miRNA promoter was expressed (Author response image 2, Table 7). The activity of each miRNA promoter was then determined by luciferase reporter assay. (HOXD10 activates the miRNA promoters, whereas NFkB represses the activities.) Expression of WT NFκB decoys, but not MT decoys, derepressed the activities of all the 3 miRNA promoters in Ctrl cells, suggesting that WT decoys sequestered NFκB, but MT decoys did not (Figure 4—figure supplement 1). Similarly, expression of WT HOXD10 decoys, but not MT decoys, inhibited the activities of all the 3 miRNA promoters in HOXD10-overexpressing cells, suggesting that WT decoys sequestered HOXD10 while MT decoys did not (Figure 4—figure supplement 1). These results clearly confirm that NFκB and HOXD10 indeed bind the predicted sequences in the promoters of the three miRNAs, which was abrogated by point mutations in these sequences.

**Author response image 2. respfig2:** Scheme of expressing of NFkB or HOXD10 decoy that sequesters the transcription factor. (A) Parental T4-2 cells have a high level of NFkB and low level of HOXD10. These two transcription factors are proposed to antagonize each other. Binding of NFkB to the miRNA promoter inhibits miRNA expression. (B) T4-2 cells will be transfected with the wild-type NFkB binding sequence decoy, which will sequester NFkB. This will inhibit NFkB binding to the miRNA promoter, but promote HOXD10 binding to the promoter, leading to miRNA expression. (C) In contrast, expression of the mutant NFkB binding sequence decoy will not sequester NFkB, and miRNA expression will remain inhibited. (D) T4-2 overexpressing HOXD10 cDNA promotes its binding to the miRNA promoter, elevating the expression of the miRNA. (E) T4-2 cells overexpressing HOXD10 will be transfected with the wild-type HOXD10 binding sequence decoy, which will sequester HOXD10. This will promote NFkB binding to the promoter and inhibits miRNA expression. (F) In contrast, expression of the mutant decoy in T4-2 cells overexpressing HOXD10 will not sequester HOXD10. HOXD10 will stay bind to the miRNA promoter, promoting miRNA expression.

A related critical issue is whether these observed TF motifs are significantly enriched in putative target promoter regions compared to genomic background where these sequences could be frequently found. Similarly in the P53 regulation of laminin transcription experiment: is the occurrence of P53 motifs enriched over genomic background?

We appreciate the constructive questions. In the revised manuscript, we have included the detailed information of these predicted transcription factor (TF) binding sites within the promoter regions (as predicted by AlGGEN PROMO software). This is shown in (Table 6 and Table 8). The feasibility of these predicted sites is indicated as the ‘Dissimilarity’ to the canonical sequence (0% as the best match). The significance of the predicted site is indicated as the “Frequency” in the genomic background (‘Random Expectancy’ (RE) value x 10-3)(5). Table 6 and Table 8 clearly show that these predicted sites are highly likely (low Dissimilarity and Low Frequency in the genome). Note that RE is calculated using a model where all the 4 nucleotides (A, T, G and C) have the equal frequency (equally) or a model where the 4 nucleotides have the same frequency as the query sequence (miR-34c: 1-0 kb; miR-30e: 3-2 kb; miR-144: 3-2 kb; LAMA3: -4.5 ~ +0.5 kb from transcription start site (TSS))(5).

3) The Introduction extensively describes the results. This is somewhat repetitive with the Results section. Revision of the Introduction would focus readers to the important concepts.

We appreciate the reviewer’s suggestion. We have re-written the Introduction section and removed the paragraphs that were repetitive with the Results sections.

References:

1) Fleming I, Hecker M, Busse R. Intracellular alkalinization induced by bradykinin sustains activation of the constitutive nitric oxide synthase in endothelial cells. Circ Res. 1994;74(6):1220-6.

2) Mas M. A Close Look at the Endothelium: Its Role in the Regulation of Vasomotor Tone. Euro Urol Suppl 2009;8:48-57.

3) Rabender CS, Alam A, Sundaresan G, Cardnell RJ, Yakovlev VA, Mukhopadhyay ND, Graves P, Zweit J, Mikkelsen RB. The Role of Nitric Oxide Synthase Uncoupling in Tumor Progression. Mol Cancer Res. 2015;13(6):1034-43.

4) Osako MK, Nakagami H, Morishita R. Chapter: Nucleic Acid Drugs, Development and Modification of Decoy Oligodeoxynucleotides for Clinical Application. Advances in Polymer Science: Springer-Verlag Berlin Heidelberg 2011. p. 49-59.

5) Farré D, Roset R, Huerta M, Adsuara JE, Roselló L, Albà MM, Messeguer X. Identification of patterns in biological sequences at the ALGGEN server: PROMO and MALGEN. Nucleic Acids Res. 2003;31(13):3651-3.